# What Can an LLM Flip if It Fails to Flip Coins?

## Abstract

Large language models (LLMs) can often accurately describe probability distributions using natural language, yet they still struggle to generate faithful samples from them. This mismatch limits their use in tasks requiring reliable stochasticity, such as Monte Carlo methods, agent-based simulations, and randomized decision-making. We investigate this gap between knowledge and sampling in the context of Bernoulli distributions. We introduce Verbalized Rejection Sampling (VRS), a natural-language adaptation of classical rejection sampling that prompts the LLM to reason about and accept or reject proposed samples. Despite relying on the same Bernoulli mechanism internally, VRS substantially reduces sampling bias across models. We provide a theoretical analysis showing that, under mild assumptions, VRS improves over direct sampling, with gains attributable to both the algorithm and prompt design. More broadly, our results show how classical probabilistic tools can be verbalized and embedded into LLM workflows to improve reliability, without requiring access to model internals or heavy prompt engineering.

## 1 Introduction

Large language models (LLMs) have demonstrated remarkable capabilities in generating coherent text and even performing reasoning tasks. An emerging question is whether LLMs can understand and reproduce probabilistic processes when prompted in natural language. In particular, if we ask an LLM to behave like a random sampler for a known distribution (e.g., produce coin flip outcomes with a given probability), will it faithfully do so? Reliable sampling underpins Monte Carlo algorithms [13, 19], probabilistic programming [4], agent-based simulations [11, 3], and randomized decision making [16, 15]; yet, despite randomness being central to modern computation, the extent to which contemporary LLMs can generate faithful i.i.d. samples remains largely unexplored.

Recent work has begun to study LLMs not just as next-word predictors but as generators of random outcomes drawn from specified distributions. Empirical evidence shows that, while LLMs can infer probability distributions [6] and do Bayesian updates to approximately infer a coin's bias when given data [7], their own samples from a distribution remain biased [11]. Figure 1(a;b) illustrate this gap for Bernoulli distributions. Hence, LLMs know what a fair coin is, but they struggle to behave like one.

This mismatch poses concrete risks from a user's perspective. A user who sees an LLM accurately reasoning about a distribution might trust it to sample from that distribution; hidden bias can then contaminate downstream workflows, skew survey simulators, or introduce unfairness in stochastic tie-breakers. If an LLM cannot flip a fair coin, could it be trusted to sample from more complex distributions? This raises safety, reliability, and fairness concerns across the stack.

In the setting of Bernoulli distributions, we present a comprehensive study of correcting LLM sampling bias via a language-adapted rejection-sampling framework, and uncover surprising interactions between prompt design and algorithmic guarantees. Our contributions include:

- **Sampling Faithfulness Study (Section 4).** We measure how faithfully LLMs generate i.i.d. Bernoulli samples when prompted directly. Across four models, sampling bias varies significantly with the phrasing of the distribution. *Chain-of-thought does not guarantee improvement.* We also quantify the gap between a model's ability to identify a distribution and its ability to simulate it.

- **Verbalized Rejection Sampling (VRS) (Section 5).** We adapt the classical rejection sampling method through natural language into LLMs. VRS is model-agnostic (for both open-source and proprietary LLMs), requires no access to the model weights, and keeps the LLM in a black-box.

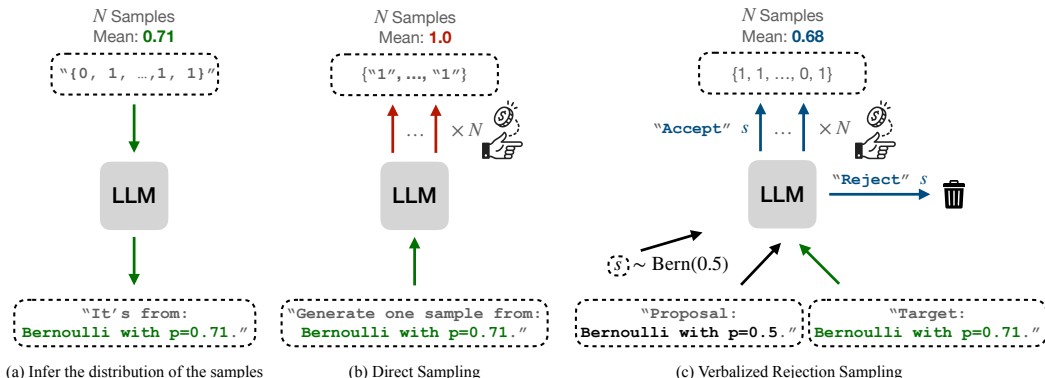

Figure 1: Illustrations of the knowledge-sampling gap and two different sampling methods.

Given a fixed prompt with textual descriptions of the target and proposal distributions alongside a candidate sample, the LLM is instructed to perform the accept/reject step. Our empirical study shows a significant reduction of the bias for the samples.

- **Empirical and Theoretical Insights (Section 6).** Effectively, VRS draws a Bernoulli random variable to decide whether to accept a proposed sample. Counter-intuitively, this indirection produces less sampling bias than prompting the model to output a sample directly. We analyze this phenomenon theoretically, proving (under mild assumptions) that VRS can generate samples with less bias than direct sampling and separating the gains attributable to the prompt phrasing from those guaranteed by the algorithm itself.

Beyond correcting the specific failure mode of Bernoulli sampling, our study opens a broader path towards integrating principled randomness into LLM-based systems. Faithful Bernoulli generation is a basic requirement for reliable LLM-driven simulations and stochastic reasoning. Our results show that a lightweight, theoretically sound wrapper—without model access or hyper-parameter tuning—substantially narrows the knowledge-sampling gap. More broadly, our work illustrates how classical statistical tools can be verbalized and paired with LLMs to deliver reliability without resorting to opaque prompt engineering.

## 2 RELATED WORK

**Sampling and flipping coins with LLMs.** Recent work shows that LLMs often exhibit a gap between knowing and sampling from a distribution. For example, LLMs can describe the target probabilities, yet when asked to "roll a die" or "flip a coin" their outputs exhibit large bias. Incorporating code generation with Python tool use can alleviate the problem [6]. In contrast, we focus on improving sampling within the natural language space, leveraging LLMs' inherent probabilistic reasoning capabilities. While one could bypass the model to obtain true samples from a target distribution, enabling LLMs to faithfully perform such tasks themselves is both practically useful and scientifically insightful. Also, when LLMs are asked to "flip a fair coin" and "flip 20 fair coins", they not only replicate human biases but often amplify them [14]. Another work probes the online learning setting of Bernoulli distribution from a Bayesian inference angle [7], showing that with sufficient in-context examples, LLMs update their estimate of a coin's bias roughly following Bayes' rule. Unlike their focus on online learning and belief updating, we do not assume sequential access to data and instead concentrate on the generation of i.i.d. samples from a fixed Bernoulli distribution. Similar gaps exist in settings beyond Bernoulli (e.g., poll simulation, categorical distribution), showing that LLMs can summarize distributions but fail to sample from them reliably, echoing the Bernoulli findings on a higher-dimensional setup [11, 8]. Together, these studies reveal a recurring pattern: LLMs know the right distributions but struggle to sample from them faithfully. Our work aims to reduce this mismatch by adapting the rejection sampling algorithm to LLMs, leveraging their internal probabilistic behavior to guide natural language based sampling.

**Natural language and text based parameterization.** Recent work explores using natural language to parameterize models, treating LLMs as inference engines that interpret and evaluate these descrip-

Figure 2: Prompt templates for direct sampling and Verbalized Rejection Sampling.

tions. This makes model specification more accessible and interpretable. LLM Processes [12], where LLMs generate predictive distributions conditioned on natural language inputs and in-context data, operates in an in-context, non-parametric style and requires access to token logits. In contrast, we treat language as a parametric description of a fixed distribution, without past data or logit access. In Verbalized Machine Learning (VML; [18]), prompts are treated as natural language parameters for deterministic functions. Our work instead focuses on probabilistic distributions and faithful sampling. Additionally, new theoretical frameworks demonstrating that a finite set of function compositions, analogous to a vocabulary, can approximate any continuous mapping, drawing parallels between linguistic compositionality and function approximation [2]. These studies underscore the potential of natural language as a medium for specifying probabilistic models. In our work, we focus on the Bernoulli distribution as a fundamental case study, demonstrating how LLMs can be guided to generate faithful samples from a simple yet foundational probabilistic model.

## 3 PROBLEM SETUP

Our investigation focuses on the ability of LLMs to generate faithful i.i.d. samples from distributions described purely in natural language. Focusing on Bernoulli distributions, defined by a single numerical parameter $p \in [0, 1]$, we treat LLMs as samplers accessed solely through text interaction.

### 3.1 PARAMETERIZING DISTRIBUTIONS IN NATURAL LANGUAGE

In our setting, the distribution is parameterized by a textual prompt. Formally, we denote this natural language parameterized distribution as $P(x; \theta)$, where $\theta$ captures both the underlying numerical parameter $p$ and the linguistic phrasing of the prompt. Figure 2(left) shows an example, where

$P(x; \theta)$ = *"Your task is to generate a sample from the set {0, 1}, where the probability of 1 is 0.0."*.

For the same $p$, different phrasings may lead to different sampling behaviors. We test several ways of phrasing a Bernoulli distribution, and write $P(x; p)$ for a fix phrasing. For each phrasing, we test 101 values of $p \in \{0.0, 0.01, 0.02, \ldots, 1.0\}$. For each $p$, we query the LLM 100 times independently with the same prompt, and extract the binary output (i.e., '0' or '1') to form the resulting i.i.d. samples.

### 3.2 LLMS AS BLACK-BOX SAMPLERS

We treat LLMs as black-box samplers, accessed solely via APIs. The only controllable input is the prompt; the only observable output is text. For open-source models, we use vLLM [9], but we assume no access to internals such as weights, activations, or token-level logits. This contrasts with prior work [7, 8, 12] that uses output token logits to estimate sampling probabilities.

This API-only setup allows consistent evaluation across both open-source and proprietary models, reflecting realistic usage where internals are inaccessible. It also better supports techniques like

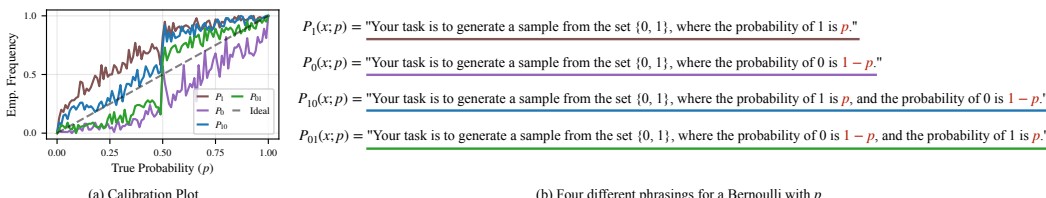

(a) Calibration Plot          (b) Four different phrasings for a Bernoulli with $p$

Figure 4: Calibration plots for direct sampling and the four different phrasings.

chain-of-thought (CoT; [17]) prompting, which can distort token-level probabilities by conditioning on generated reasoning: with CoT, logits reflect $p(x \mid \text{reasoning for } x)$ instead of the intended $p(x)$. We also fix all decoding hyperparameters (e.g., temperature, top-k) to their default values given in the API, since most real world users do not adjust them, and often do not have the ability to do so.

## 4   How Reliable is Direct Sampling?

This section examines the reliability of direct sampling from LLMs. We first compare their ability to generate samples to their ability to recognize distributions, then explore how prompt phrasing affects sampling bias, and finally test whether chain-of-thought reasoning improves sample quality.

### 4.1   Measuring the Knowledge-Sampling Gap

To assess the gap between an LLM's understanding of a Bernoulli distribution and its ability to sample from it, we compare its evaluative and generative performance in a controlled setup, using Llama-3.1-70B-Instruct [5]. We first test the model's ability to identify the correct Bernoulli distribution from data. For 11 equally spaced probabilities $p_0, ..., p_{10}$, s.t. $p_i \in [0, 1]$, we generate 100 i.i.d. samples using Python, forming datasets $S_i$. For each pair $(i, j)$, we prompt the LLM to decide whether $S_i$ was drawn from $\text{Bern}(p_j)$, producing an $11 \times 11$ response matrix. Diagonal entries should be "Yes", off-diagonals "No". We repeat this process five times and report average accuracies in Figure 3(a). We then test the model's sampling behavior by prompting it to generate 100 samples for each $p_i$, using the template in Figure 2(left). The resulting sets $\hat{S}_i$ are evaluated using the same method as before. The average accuracies over five runs are reported in in Figure 3(b).

The left panel shows high off-diagonal accuracy for Python generated data (i.e., confidently rejecting incorrect hypotheses), with minor errors along the diagonal due to natural sample variation (e.g., 48 ones out of 100 for $p = 0.5$ may lead to confusion with $p = 0.48$, hence, rejecting the correct hypotheses). In contrast, the right panel shows major degradation for LLM-generated samples. Diagonal accuracy drops significantly for all $p_i$, except the edge cases when $p = 0.0$ and $p = 1.0$. Moreover, we observe an asymmetry in the off-diagonal en-

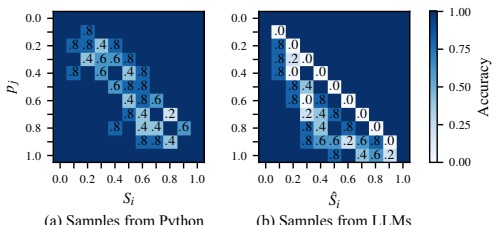

(a) Samples from Python    (b) Samples from LLMs

Figure 3: Recognition accuracy matrix.

tries: the lower triangle of the matrix exhibits much worse accuracy than the upper triangle. This indicates that samples from $p_i$ are often misclassified as having come from $p_j$ with $j > i$, suggesting that the LLM-generated samples are consistently biased toward ones. These results reveal a clear knowledge–sampling gap: LLMs can evaluate distributions well but fail to sample from them faithfully. Unlike question answering, where each input has a correct target, i.i.d. sampling lacks per-instance ground truth, making it a fundamentally different and underexplored capability.

### 4.2   How Much Can Prompt Phrasing Reduce Sampling Bias?

The previous section used a single fixed phrasing to describe the Bernoulli distribution (see Figure 2, left). Yet, natural language allows many equivalent ways to express the same distribution, raising the question: how much can phrasing affect sampling bias? In the prior setup, the prompt emphasized the

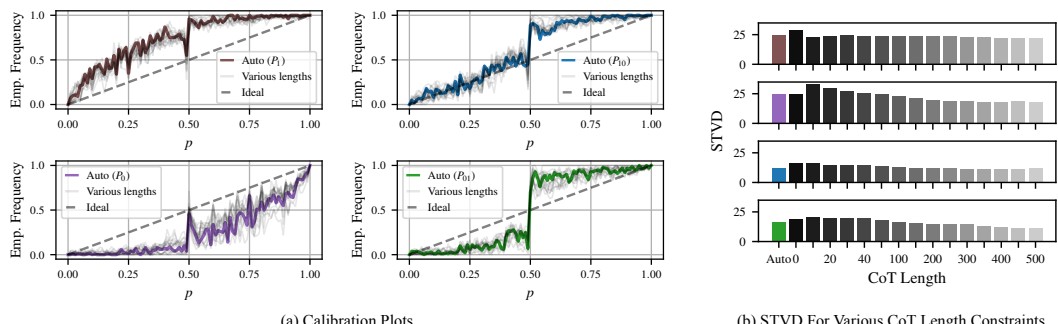

(a) Calibration Plots        (b) STVD For Various CoT Length Constraints

Figure 5: Calibration plots and STVD trend for various reasoning length constraints.

probability of generating a 1, denoted $P_1(x; p)$, as illustrated in Figure 4(b). Notably, this formulation focuses solely on the probability of generating a 1, which may partly explain the tendency of the model to produce more 1s than 0s in the in the sampled outputs.

To explore this, we test three alternative phrasings that shift or balance the focus across outcomes, as shown in Figure 4(b). For each, we sample across a range of $p$ values using Llama-3.1 and plot the empirical frequency of 1s against the ground truth, yielding calibration curves shown in Figure 4(a). The calibration curves show that the balanced descriptions, i.e., those stating both probabilities, yield samples that are better calibrated. Nevertheless, all four phrasings result in noticeable bias. This result resonates with [1], where they found that when prompting humans to imagine a coin flip, mentioning only 'heads' or mentioning only 'tails' will lead to a similar sampling bias.

**Quantitative comparison using Sum of TV Distance (STVD).** To quantify the calibration performance of different phrasings, we compute the area between each calibration curve and the ideal diagonal reference line. Specifically, for each $p_i$, we calculate the absolute difference between the empirical sampling frequency $\tilde{p}_i$ and the true value $p_i$, and sum these over all 101 values, i.e., $\text{STVD} = \sum_{i=0}^{100} |\tilde{p}_i - p_i|$. Since this absolute difference corresponds to the total variation (TV) distance between two Bernoulli distributions, we refer to the resulting metric as the Sum of TV Distances (STVD) where smaller is better. See Appendix A.1 for more details about the TV distance.

Table 1 presents the STVD values for the four phrasings under direct sampling. For Llama 3.1, the best-performing phrasing $P_{10}$ achieves an STVD of 12.50, nearly half that of the baseline $P_1$, which scores 25.36. We also include results for other LLMs, including GPT-4.1-nano, DeepSeekV3 [10], and Qwen-2.5 72B [20]. Interestingly, the best-performing phrasing varies across models, as highlighted by the underlined entries. The calibration plots for the other models can be found in Appendix B.1.

These findings suggest that while prompt design can influence sampling bias, relying solely on prompt engineering to eliminate bias can be difficult and inconsistent across model family, and additional mechanisms are likely needed for more systematic approaches to correct sampling bias.

Table 1: Quantitative comparison between Direct Sampling and VRS in STVD ($\downarrow$).

| Method | Llama-3.1 70B | | | | | GPT-4.1-nano | | | | | DeepSeekV3 | | | | | Qwen-2.5 72B | | | | |
|---|---|---|---|---|---|---|---|---|---|---|---|---|---|---|---|---|---|---|---|---|
| | $P_1$ | $P_0$ | $P_{10}$ | $P_{01}$ | mean | $P_1$ | $P_0$ | $P_{10}$ | $P_{01}$ | mean | $P_1$ | $P_0$ | $P_{10}$ | $P_{01}$ | mean | $P_1$ | $P_0$ | $P_{10}$ | $P_{01}$ | mean |
| Direct | 25.36 | 24.79 | _12.50_ | 16.59 | 19.81 | 17.87 | 30.23 | _16.63_ | 19.24 | 21.00 | _17.76_ | 19.39 | 20.78 | 23.26 | 20.30 | 20.73 | _18.72_ | 19.00 | 22.64 | 20.27 |
| VRS | 5.73 | 7.64 | _5.36_ | 5.60 | **6.08** | 12.96 | 13.06 | 9.50 | _8.46_ | **11.00** | 5.34 | 9.06 | _5.29_ | 6.94 | **6.66** | 5.93 | 6.35 | _4.49_ | 5.12 | **5.47** |

### 4.3 DOES CHAIN-OF-THOUGHT (CoT) HELP SAMPLING?

Since phrasing alone does not eliminate sampling bias, we explore whether modifying the instruction for the output can help. Prior work [14, 7, 8, 12] often asks LLMs to output the sample immediately, enabling access to token logits for estimating predictive distributions. However, this approach is constrained to open-source models and treats LLMs more as likelihood models than samplers. In our setting, we only use LLMs for sampling and do not require access to logits or early output. This

allows us to apply CoT [17] prompting, where the model first generates reasoning before giving its final answer. While sampling differs from question answering, CoT may increase output variability by encouraging diverse reasoning paths, potentially reducing bias.

To test this, we instruct the model to produce reasoning of varying lengths $N$ (ranging from 0 to 500 words) before answering, along to an 'Auto' setting where no length constraint is imposed. The 'Auto' is the default setting for experiments in previous sections, which uses the template in Figure 2(left). For different $N$, we modify the '*Explanations*' instruction in the prompt template to include a sentence saying that '*Your analysis must have around N words*'.

Figure 5 presents the calibration plots (left) and STVD scores (right) for Llama-3.1 under different CoT length constraints. Overall, reasoning length has limited effect on bias, though longer CoT slightly improves calibration. Direct output without reasoning often performs worse than the 'Auto' setting. However, this pattern does not hold across models. As shown in Figure 6, GPT-4.1 and Qwen2.5 show no consistent improvement with longer CoT; in some cases, STVD increases as reasoning length grows. These mixed results suggest that, unlike in question answering, CoT is not a reliable method for reducing sampling bias, and its effect is model-dependent. For consistency, we use 'Auto' in all remaining experiments.

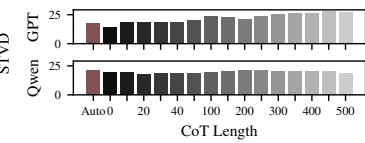

Figure 6: STVD vs CoT Length.

# 5 VERBALIZED REJECTION SAMPLING

In the previous section, we explored ways to reduce sampling bias through prompt phrasing and instruction design. While these strategies do influence the behavior of LLMs, the results suggest that prompt-only interventions are insufficient for reliably eliminating bias. If direct sampling cannot be fully corrected through language alone, we may instead embrace the bias and mitigate it using algorithmic techniques. In probabilistic methods, several algorithms exist to transform biased proposals into unbiased samples. One such method is rejection sampling, which generates candidate samples from a proposal distribution and selectively accepts them to match a desired target distribution. In the remainder of this section, we adapt rejection sampling to operate entirely within the language interface of LLMs, and we refer to this method as verbalized rejection sampling (VRS).

## 5.1 REJECTION SAMPLING

Rejection sampling is a sampling technique to generate samples from a target distribution $P$ while only having access to samples from a (typically simpler) proposal distribution $Q$. We assume that both $P$ and $Q$ can be evaluated (but only $Q$ can be directly sampled from). The general idea is that we can generate a sample from $P$ by instead sampling from $Q$ and accepting the sample with probability $P(x)/(MQ(x))$ where $M < \infty$ is a bound on the ratio $P(x)/Q(x)$. We assume that both $P$ and $Q$ are Bernoulli distributions with parameters $p$ and $q$. In this case, we can compute $M$ analytically as: $M = \max\{p/q, (1-p)/(1-q)\}$. Let $A(x)$ denote the acceptance probability of $x \sim Q$ which is

$$A(x) = \begin{cases} \frac{P(x)}{MQ(x)} = \frac{p}{Mq} & \text{if } x = 1 \\ \frac{P(x)}{MQ(x)} = \frac{1-p}{M(1-q)} & \text{if } x = 0 \end{cases}. \tag{1}$$

The accept/reject step effectively draws a sample from $\text{Bern}(A(x))$. The overall acceptance rate is $\alpha = \sum_{x \in \{0,1\}} Q(x)A(x) = 1/M$. See Appendix A.2 for more details about rejection sampling.

## 5.2 ADAPTING REJECTION SAMPLING TO LLMS

Figure 1(c) illustrates the overall idea behind VRS. Classical rejection sampling requires three inputs: the target distribution $P$, the proposal distribution $Q$, and a sample $x \sim Q$. The algorithm evaluates whether to accept or reject $x$ based on these inputs, returning a binary decision. To implement this in the LLM setting, we design a prompt template (Figure 2, right) that verbalizes all three components, i.e., descriptions of $P, Q$, and the proposed sample $x$, as natural language. These are inserted into fixed slots in the template. The model is instructed to reason through its decision and then output a single letter from $\{T, F\}$, indicating whether to accept (T) or reject (F) the sample. We send

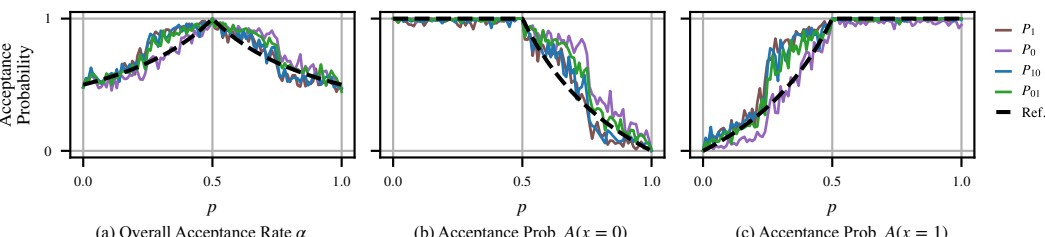

Figure 8: Empirical acceptance rates for VRS.

the completed prompt to the LLM and parse its response. If the response indicates acceptance, we retain the sample; otherwise, we generate a new proposed sample and repeat the process. This loop continues until we collect the required number of accepted samples (pseudocode in Appendix D.1).

### 5.3 Experiments

We evaluate VRS on four different LLMs: Llama-3.1, GPT-4.1-nano, DeepSeekV3, and Qwen-2.5. For each model, we run VRS until it accepts 100 samples for each of the 101 values of $p \in [0.0, 1.0]$, following the same setup as in the direct sampling experiments. As the proposal distribution $Q$, we fix it to a uniform Bernoulli with $q = 0.5$ across all values of $p$. The resulting calibration plot for Llama-3.1 is shown in Figure 7, and the corresponding STVD scores across all models are included in Table 1. The calibration plots for other three LLMs can be found in Appendix B.2.

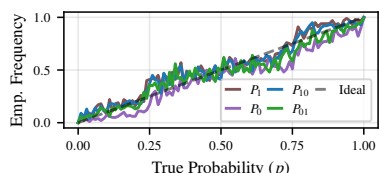

Figure 7: Calibration plot for VRS

Comparing the calibration plot for VRS (Figure 7) with that of direct sampling (Figure 4a), we observe a significant reduction in sampling bias. Across all four prompt phrasings, the calibration curves under VRS closely align with the ideal diagonal reference, indicating much improved fidelity to the target Bernoulli distributions. Figure 8 shows the corresponding empirical acceptance probabilities, which seem to align well with the analytical targets. The improvement is also reflected quantitatively in Table 1: the STVD scores for VRS are substantially lower than those for direct sampling, with most cases showing a reduction of over 50%. In some instances, STVD drops to nearly 25% of the original value. Crucially, this improvement holds across all four LLMs tested (i.e., Llama-3.1, GPT-4.1-nano, DeepSeekV3, and Qwen-2.5), demonstrating that VRS consistently mitigates bias and does so independently of the underlying model.

## 6 Why Does Verbalized Rejection Sampling Work?

The effectiveness of VRS in reducing sampling bias is surprising at first glance since, internally, VRS still relies on the LLM to perform a Bernoulli trial, i.e., deciding whether to accept or reject a sample, which is precisely the type of stochastic behavior we have shown LLMs to struggle with.

*If LLMs are biased in direct sampling, why does wrapping the decision in rejection sampling help?*

Is the improved calibration a result of the specific prompt design used in VRS? Or does the rejection sampling algorithm itself introduce structural guarantees that correct bias, even when implemented via a biased LLM? The remainder of this section explores these possibilities empirically and theoretically.

### 6.1 Is the Magic in the Prompt?

To investigate whether VRS's improvement stems purely from prompt design, we remove external randomness by fixing the proposed sample to a constant, i.e., $x = 1$. In this case, a faithful LLM should accept with probability $A(1)$, as defined in Equation (1). We compare this with the empirical acceptance probability $\tilde{A}(1)$, estimated from the LLM's responses. Figure 8(c) shows $\tilde{A}(1)$ for various $p$, using a fixed proposal $Q = \text{Bern}(0.5)$. For the trivial case $p > 0.5$, the alignment is strong.

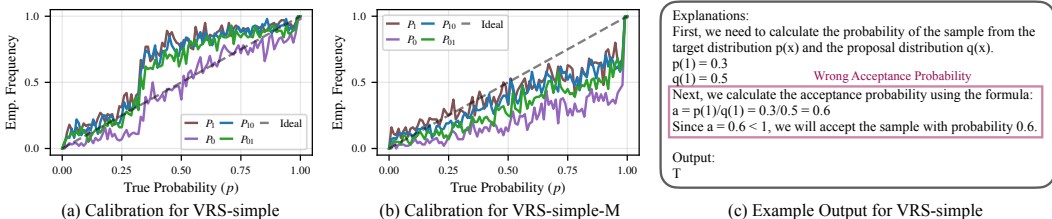

(a) Calibration for VRS-simple  (b) Calibration for VRS-simple-M  (c) Example Output for VRS-simple

Figure 9: Calibration plots for two ablations and an example LLMs output for VRS-simple.

For $p < 0.5$, the results appear reasonable overall but show a consistent bias, particularly in the range $p \in [0.2, 0.5]$. To compare more directly with direct sampling, we evaluate $\tilde{A}(1)$ over 101 equally spaced values of $A(1)$, using the inverse of Equation (1) to recover the corresponding $p$. For each, we generate a VRS prompt with the computed $p$, a fixed $Q = \text{Bern}(0.5)$, and a fixed sample $x = 1$. We refer to this setup with fixed proposal and no introduced randomness as *VRS-simple*. If prompt design alone explains the improvement, VRS-simple should outperform direct sampling in calibration.

Figure 9(a) shows the calibration plot for VRS-simple using Llama-3.1. Compared to direct sampling (Figure 4a), the results are slightly more calibrated. Table 2 confirms this, with the mean STVD dropping from 19.81 to 11.86 (see Appendix B.4 for the full table). This suggests the VRS prompt helps reduce bias for direct sampling. However, VRS-simple relies on explicitly computing the inverse of Equation (1) to tailor the prompt to each target $p$, and the improvement remains modest compared to full VRS.

Table 2: Ablation STVD ($\downarrow$)

| Method | mean (over 5 runs) |
|---|---|
| Direct | $19.81 \pm 0.15$ |
| VRS | $6.08 \pm 0.12$ |
| VRS-simple | $11.86 \pm 0.13$ |
| VRS-simple-M | $18.45 \pm 0.35$ |
| VRS-M | $7.36 \pm 0.14$ |

**Magic or Mirage?**  To further understand why the VRS prompt improves sampling, we examine whether its structure encourages the model to reason differently. One hypothesis is that phrasing the sampling task in the context of rejection sampling prompts the LLM to internally compute acceptance probabilities, potentially disrupting its default biases learned during pretraining. To test this, we manually analyzed the model's reasoning outputs from VRS-simple (see Figure 9(c)). We found that, while the model often tries to derive the acceptance probability, it frequently does so incorrectly. In the non-trivial cases where $A(x) \neq 1$, the model tends to compute only the ratio $P(x)/Q(x)$, omitting the constant $M$ in the denominator.

*Could this incorrect derivation be the reason behind the improvement?*  To test that, we designed variants of VRS-simple and VRS where we explicitly instruct the model to compute and use $M$ correctly. We refer to these as VRS-simple-M and VRS-M, respectively. The calibration plot for VRS-simple-M is shown in Figure 9(b), with corresponding STVD scores in Table 2. Through output inspection, we verified that the LLM now correctly computes the constant $M$ in its reasoning. The correction in VRS-simple-M leads to slightly better performance from 19.81 to 18.45. However, for the full VRS setup, adding the $M$-instruction results in a slight degradation, with STVD rising from 6.08 to 7.36, though still outperforming direct sampling.

These results suggest that the improvement from the VRS prompt is not due to accurate computation of the acceptance probability. Instead, the prompt seems to help in an unexpected way, but it alone cannot explain the full benefit. The remaining gains likely come from the rejection sampling mechanism itself, rather than prompt phrasing alone.

## 6.2 IS THE IMPROVEMENT FROM THE ALGORITHM?

Prompt design alone cannot fully explain the gains from VRS. To analyze the role of the algorithm itself, we model the LLM as a biased Bernoulli sampler. In VRS, this means the acceptance decision is not sampled from the true probability $A(x)$, but from a perturbed version $\tilde{A}(x) = A(x) + e(x)$, where $e(x)$ represents the model's bias. Based on this, we can derive the following proposition.

**Proposition 1.** *[Informal, see Proposition 1 in Appendix A.3.] Let $P$ and $Q$ be Bernoulli distributions (target and proposal with parameters $p$ and $q$, respectively). Let $\tilde{P}$ denote the distribution resulting from rejection sampling with acceptance probability $A(x) + e(x)$, and assume a bound on the model's*

*bias, i.e., $|e(x)| \leq c \in \mathbb{R}$. Then, with $M$ defined in Section 5.1, we have*

$$\mathrm{TV}(\tilde{P}, P) \leq \frac{Mc}{1 - Mc}. \tag{2}$$

Intuitively, $M$ implies how "different" the proposal $Q$ is from the target $P$ and $c$ bounds the bias in the acceptance step. For $c = 0$, we recover rejection sampling with $\mathrm{TV}(\tilde{P}, P) = 0$. Otherwise, for a fixed $c > 0$, $\mathrm{TV}(\tilde{P}, P)$ grows quadratically with $M$.

From empirical observations (see Figure 8(b;c)) we note that $c \approx 0$ when the acceptance probability is trivial, i.e., for $A(x) = 1$. We can integrate this assumption and get the following tighter bound.

**Proposition 2.** *[Informal, see Proposition 2 in Appendix A.4.] Following Proposition 1 but with the additional assumption that $A(\cdot)$ is only biased in the non-trivial case, i.e., $\tilde{A}(\hat{x}) = A(\hat{x}) + e(\hat{x})$ if $A(\hat{x}) < 1$ (whereas $\tilde{A}(x^*) = A(x^*)$ if $A(x^*) = 1$), we have*

$$\mathrm{TV}(\tilde{P}, P) \leq \frac{Q(\hat{x})Mc}{(1 - Q(\hat{x})Mc)}. \tag{3}$$

Out of the two possible events for $x$, we use $\hat{x}$ to refer to the event that achieves non-trivial $A(\hat{x}) < 1$. Intuitively, $Q(\hat{x})$ "damps" the error as $Q(\hat{x}) \leq 1$, which results in a tighter bound.

Let $\bar{P}$ denote the distribution resulting from direct sampling with the same bias $e(x)$. We can now derive when VRS ($\tilde{P}$) is better than direct sampling ($\bar{P}$). We get the following corollary.

**Corollary 1.** *[Informal, see Corollary 1 in Appendix A.5.] Following Proposition 2 and assuming that $\bar{P}$ has the same bias as in $\tilde{A}(x)$, i.e., $\bar{p} = p + e, e \leq |c|$, then (with $\alpha = 1/M$, see Section 5.1)*

$$\mathrm{TV}(\tilde{P}, P) < \mathrm{TV}(\bar{P}, P) \overset{(i)}{\Longleftrightarrow} \frac{Q(\hat{x})}{\alpha}(1 + c) \leq 1 \overset{(ii)}{\Longleftrightarrow} c < \frac{1}{Q(\hat{x})M} - 1. \tag{4}$$

This implies: $(i)$ VRS is better than direct sampling if the biased event ($\hat{x}$) is rare, i.e., if $Q(\hat{x})$ is smaller than the acceptance rate $\alpha$; $(ii)$ equivalently, this can be translated into a bound on $c$, which is maximized when $Q$ and $P$ are similar, i.e., if $M \to 1$. Otherwise, direct sampling is better than VRS.

In our experiments we fix the proposal to $q = 0.5$. This allows us, for each $p$, to compute the constants $M$ and $c$ (i.e., the upper bound on $|e(x)|$) under which VRS outperforms direct sampling. Figure 10 shows the calibration plot of Figure 4(a) and visualizes a shaded black box of $p$ plus and minus $c$, i.e., clip($p \pm c, 0.0, 1.0$). The box is largest (vertically) if $P$ and $Q$ are similar (i.e., $M \to 1$, see $(ii)$ above). In most cases, the empirical frequencies of direct sampling fall within this box, satisfying condition $(ii)$. This provides strong evidence that the primary source of VRS's improvement comes from the rejection sampling algorithm and not from prompt effects.

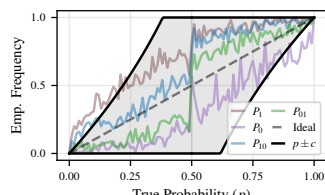

Figure 10: Calibration plots with error bounds $\pm c$ overlaid.

# 7 CONCLUSION

We examined the ability of LLMs to sample from natural-language-described distributions, using Bernoulli as a test case. While LLMs can evaluate whether data matches a distribution, they struggle to generate unbiased samples, revealing a clear knowledge-sampling gap. This highlights that *sampling is a fundamentally distinct ability from question answering*: evaluation tasks have clear supervision, while i.i.d. sampling lacks per-instance ground truth and is only verifiable at the distribution level. We showed that prompt phrasing or chain-of-thought reasoning could not guarantee improvement. To address this, we proposed Verbalized Rejection Sampling (VRS), a lightweight adaptation of classical rejection sampling expressed entirely in natural language. VRS improves calibration across models without accessing logits or tuning decoding parameters, and our analysis shows that the algorithm, not just prompt design, is key to its success. Although our main analysis is for Bernoulli (which has been widely recognized as a foundational testbed for assessing LLM sampling behavior [6, 14, 7]), we observed the effectiveness of VRS also in Binomial distributions (see Appendix C), demonstrating the potential that the framework can be adapted to more complex families. Beyond correcting this specific failure mode, our work points to a broader path: integrating principled randomness into LLM-based systems. VRS illustrates how classical probabilistic tools can be verbalized and embedded into LLM workflows to improve reliability without relying on opaque prompt engineering.

THE USE OF LARGE LANGUAGE MODELS (LLMS)

We used LLMs to aid and polish writing, but they did not contribute significantly at the level of a contributing author. Our research question is also about LLMs, therefore, they are used in our experiments as the subject of investigation.

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

# Appendix for VRS

## Table of Contents

# A  THEORETICAL ANALYSIS FOR BIASED (REJECTION) SAMPLING FROM BERNOULLI DISTRIBUTIONS

This section is structured as follows. We introduce the total variation distance in Appendix A.1 and state the general rejection sampling problem in Appendix A.2. In Appendix A.3 we bound the worst case error assuming that the acceptance probability of the rejection sampling algorithm is biased. Appendix A.4 bounds the worst case error assuming that the model can draw exact samples if the acceptance probability is 1 and is biased in the other case. In Appendix A.5 we compare the previous bound to the error of a biased Bernoulli distribution.

## A.1  TOTAL VARIATION DISTANCE

The total variation (TV) distance measures the statistical distance between two probability distributions. We will state it below for the case where both distributions are Bernoulli.

Let $P_1$ and $P_2$ denote the probability mass functions of Bernoulli distributions with parameters $p_1$ and $p_2$, respectively. We can write the TV distance between $P_1$ and $P_2$ as

$$D_{\text{TV}}(P_1, P_2) = \frac{1}{2} \sum_{x \in \{0,1\}} |P_1(x) - P_2(x)| = |p_1 - p_2|. \tag{TV}$$

## A.2  REJECTION SAMPLING

Rejection sampling (RS) is a sampling technique to generate samples from a distribution $P$ while only having access to samples from a distribution $Q$ but assuming that both $P$ and $Q$ can be evaluated. The general idea is that we can generate a sample from $P$ by instead sampling from $Q$ and accepting the sample with probability $P(x)/(MQ(x))$ where $M < \infty$ is a bound on the ratio $P(x)/Q(x)$.

Assume both $P$ and $Q$ are Bernoulli distributions with parameters $p$ and $q$. We can compute $M$ analytically as

$$M := \max \left\{ \frac{p}{q}, \frac{1-p}{1-q} \right\}.$$

Let $A(x)$ denote the acceptance probability

$$A(x) = \begin{cases} \frac{P(x)}{MQ(x)} = \frac{p}{Mq} & \text{if } x = 1 \\ \frac{P(x)}{MQ(x)} = \frac{1-p}{M(1-q)} & \text{if } x = 0 \end{cases}.$$

Let $A$ denote the acceptance event. The unconditional acceptance probability $\mathbb{P}(A)$—called the acceptance rate $\alpha$—is the proportion of proposed samples that are accepted. It is given by

$$\alpha := \mathbb{P}\left(U \leq \frac{P(x)}{MQ(x)}\right) = \mathbb{E}\left(\frac{P(x)}{MQ(x)}\right) = \sum_{x \in \{0,1\}} Q(x)A(x) = \frac{p}{M} + \frac{1-p}{M} = \frac{1}{M}.$$

where $U \sim \text{Unif}(0,1)$. The law of the accepted samples is

$$\mathbb{P}(X = x \mid A) = \frac{\mathbb{P}(X = x, A)}{\mathbb{P}(A)} = \frac{Q(x)A(x)}{\alpha} = \frac{Q(x)\frac{P(x)}{MQ(x)}}{\alpha} = \frac{P(x)/M}{\alpha} = P(x).$$

## A.3  BIASED ACCEPTANCE PROBABILITY

We will establish a worst-case bound in terms of the TV distance for the case that the acceptance probability is biased.

**Proposition 1.** *Let $P(x), Q(x)$ be Bernoulli distributions with parameters $p$ and $q$, respectively, where $P$ is the target distribution that we want to sample from with rejection sampling and $Q(x)$ is the proposal distribution. Further, let $\tilde{P}(x)$ denote the Bernoulli distribution resulting from*

*a biased accept/reject step where we assume that the acceptance probability $\tilde{A}(x)$ is biased as $\tilde{A}(x) = A(x) + e(x)$ where $|e(x)| \leq c \in \mathbb{R}$. Then,*

$$D_{\mathrm{TV}}(\tilde{P}, P) \leq \frac{Mc}{1 - Mc}, \tag{L1}$$

*where $M = \max\{p/q, (1-p)/(1-q)\}$.*

*Proof.* Assuming a biased acceptance probability $\tilde{A}(x)$, we can split the resulting acceptance rate into

$$\tilde{\alpha} = \sum_{x \in \{0,1\}} Q(x)(A(x) + e(x)) = \underbrace{\sum_{x \in \{0,1\}} Q(x)A(x)}_{=\alpha} + \underbrace{\sum_{x \in \{0,1\}} Q(x)e(x)}_{=:\delta},$$

where $\alpha$ corresponds to the unbiased acceptance rate and $\delta$ denotes the deviation from it. We assume that $0 \leq \tilde{A}(x) \leq 1$. Note that $|\delta| \leq c$ and, therefore, $\tilde{\alpha} = \alpha + \delta \geq \alpha - c \geq 0$. Let $\tilde{A}$ denote the acceptance event. We denote the resulting law of the accepted samples by $\tilde{P}$.

$$\mathbb{P}(X = x \mid \tilde{A}) = \frac{\mathbb{P}(X = x, \tilde{A})}{\mathbb{P}(\tilde{A})} = \frac{Q(x)\tilde{A}(x)}{\tilde{\alpha}} =: \tilde{P}(x).$$

We can now upper-bound a term in the TV distance as follows.

$$|\tilde{P}(x) - P(x)| = \left| \frac{Q(x)\tilde{A}(x)}{\tilde{\alpha}} - \frac{Q(x)A(x)}{\alpha} \right| = \left| \frac{Q(x)}{\alpha\tilde{\alpha}}(\tilde{A}(x)\alpha - A(x)\tilde{\alpha}) \right|$$

$$= \left| \frac{Q(x)}{\alpha\tilde{\alpha}}((A(x) + e(x))\alpha - A(x)(\alpha + \delta)) \right| = \left| \frac{Q(x)}{\alpha\tilde{\alpha}}(e(x)\alpha - A(x)\delta)) \right|$$

$$= Q(x) \left| \frac{e(x)\alpha - A(x)\delta}{\alpha\tilde{\alpha}} \right| \tag{5}$$

$$\leq Q(x)\frac{|e(x)|\alpha + A(x)|\delta|}{\alpha(\alpha - c)} \tag{6}$$

$$\leq Q(x)\frac{c\alpha + A(x)c}{\alpha(\alpha - c)} \tag{7}$$

$$= Q(x)\frac{c}{\alpha - c}\left(1 + \frac{A(x)}{\alpha}\right)$$

In Equation (6) we used the triangle inequality, in Equation (7) we used $|e(x)| \leq c$. For the full TV distance, we get

$$D_{\mathrm{TV}}(\tilde{P}, P) = \frac{1}{2} \sum_{x \in \{0,1\}} \frac{c}{\alpha - c} Q(x)\left(1 + \frac{A(x)}{\alpha}\right) = \frac{c}{\alpha - c} = \frac{Mc}{1 - Mc}.$$

$\square$

### A.4 HALF-BIASED ACCEPTANCE PROBABILITY

In the following argument we assume that if $A(x) = 1$ there is no bias, i.e., no error ($A(x) = 1 \Rightarrow e(x) = 0$).

**Proposition 2.** *Let $P(x), Q(x)$ be Bernoulli distributions with parameters $p$ and $q$, respectively, where $P(x)$ is the target distribution that we want to sample from with rejection sampling and $Q(x)$ is the proposal distribution. Further, let $\tilde{P}(x)$ denote the Bernoulli distribution resulting from a biased accept/reject step where we assume that the acceptance probability $\tilde{A}(x)$ is biased with an additive error $e(x)$ where $|e(x)| \leq c \in \mathbb{R}$ as*

$$\tilde{A}(x) = \begin{cases} A(x) + e(x) & \text{if } A(x) < 1 \\ A(x) & \text{if } A(x) = 1 \end{cases}, \tag{M1}$$

*where $|e(x)| \leq c \in \mathbb{R}$. Then,*

$$D_{\text{TV}}(\tilde{P}, P) \leq \frac{Q(\hat{x})Mc}{(1 - Q(\hat{x})Mc)}, \tag{M2}$$

*where $M = \max\{p/q, (1-p)/(1-q)\}$ and $\hat{x}$ is chosen such that $A(\hat{x}) < 1$.*

*Proof.* Let $x^*$ be chosen such that $A(x^*) = 1 \Rightarrow e(x^*) = 0$. Let $\hat{x}$ be chosen such that $A(\hat{x}) < 1$. The resulting acceptance rate $\tilde{\alpha}$ can be states as follows.

$$\tilde{\alpha} = \sum_{x \in \{0,1\}} Q(x)(A(x) + e(x)) = \underbrace{\sum_{x \in \{0,1\}} Q(x)A(x)}_{=\alpha} + \underbrace{Q(\hat{x})e(\hat{x})}_{=:\delta}$$

We use the law $\tilde{P}$ resulting from the acceptance rate $\tilde{\alpha}$ to compute both terms, for $x^*$ and $\hat{x}$, of the TV distance. Starting form Equation (5), we get

$$|\tilde{P}(x^*) - P(x^*)| = Q(x^*)\left|\frac{e(x^*)\alpha - A(x^*)\delta}{\alpha\tilde{\alpha}}\right| = Q(x^*)\left|\frac{\delta}{\alpha\tilde{\alpha}}\right| = \frac{Q(x^*)Q(\hat{x})|c|}{\alpha\tilde{\alpha}}$$

$$\leq \frac{Q(x^*)Q(\hat{x})c}{\alpha(\alpha - Q(\hat{x})c)}$$

and

$$|\tilde{P}(\hat{x}) - P(\hat{x})| = Q(\hat{x})\left|\frac{e(\hat{x})\alpha - A(\hat{x})\delta}{\alpha\tilde{\alpha}}\right| = Q(\hat{x})|e(\hat{x})|\frac{\alpha - A(\hat{x})Q(\hat{x})}{\alpha\tilde{\alpha}}$$

$$= \frac{Q(\hat{x})(1 - Q(\hat{x}))|e(\hat{x})|}{\alpha\tilde{\alpha}} = \frac{Q(\hat{x})Q(x^*)|e(\hat{x})|}{\alpha\tilde{\alpha}}$$

$$\leq \frac{Q(x^*)Q(\hat{x})c}{\alpha(\alpha - Q(\hat{x})c)}$$

where we used the triangle-inequality in both cases. Since

$$\alpha = Q(x^*)A(x^*) + Q(\hat{x})A(\hat{x}) = Q(x^*) + Q(\hat{x})A(\hat{x}) \geq Q(x^*) = 1 - Q(\hat{x}),$$

computing the TV distance we get

$$D_{\text{TV}}(\tilde{P}, P) \leq \frac{Q(x^*)Q(\hat{x})c}{\alpha(\alpha - Q(\hat{x})c)} = \frac{(1 - Q(\hat{x}))Q(\hat{x})c}{\alpha(\alpha - Q(\hat{x})c)}$$

$$\leq \frac{\alpha Q(\hat{x})c}{\alpha(\alpha - Q(\hat{x})c)} = \frac{Q(\hat{x})c}{(\alpha - Q(\hat{x})c)}$$

$$= \frac{Q(\hat{x})Mc}{(1 - Q(\hat{x})Mc)}$$

$\square$

### A.5 COMPARISON OF HALF-BIASED SAMPLING TO DIRECT SAMPLING

In the following corollary we are comparing the distributions introduced in Appendix A.4 and biased direct sampling.

**Corollary 1.** *Let $P(x), Q(x)$ be Bernoulli distributions with parameters $p$ and $q$, respectively, where $P(x)$ is the target distribution that we want to sample from with rejection sampling and $Q(x)$ is the proposal distribution. Further, let $\tilde{P}(x)$ denote the Bernoulli distribution resulting from a biased accept/reject step where we assume that the acceptance probability $\tilde{A}(x)$ is biased with an additive error $e(x)$, where $|e(x)| \leq c \in \mathbb{R}$, if $A(x) < 1$, see Proposition 2, Equation (M1). Further, let $\bar{P}(x)$ be the Bernoulli distribution wich parameter $\bar{p}$ biased by the same additive error as*

$$\bar{p} = p + e, |e| \leq c. \tag{L1}$$

*Then, the worst-case total variation error of half-biased rejection sampling is smaller than that of direct sampling if and only if*

$$D_{\text{TV}}(\tilde{P}, P) < D_{\text{TV}}(\bar{P}, P) \iff \frac{Q(\hat{x})}{\alpha}(1 + c) \leq 1 \iff c < \frac{1}{Q(\hat{x})M} - 1, \tag{L2}$$

*where $M = \max\{p/q, (1-p)/(1-q)\}$, $\alpha = 1/M$, and $\hat{x}$ is chosen such that $A(\hat{x}) < 1$.*

*Proof.* We assume that there is an additive error $e$ when sampling with $\bar{P}$ as in Equation (L1). We can calculate the TV distance for $\bar{P}$ as

$$D_{\text{TV}}(\bar{P}, P) = |e| \leq c \tag{8}$$

Further, from Proposition 2, Equation (M2) we know that

$$D_{\text{TV}}(\tilde{P}, P) \leq \frac{Q(\hat{x})Mc}{(1 - Q(\hat{x})Mc)} \tag{9}$$

Therefore, the TV of half-biased rejection sampling (Equation (9)) to the ground truth $P$ is smaller than the TV of direct sampling (Equation (8)) if

$$\frac{Q(\hat{x})Mc}{(1 - Q(\hat{x})Mc)} < c \iff \frac{Q(\hat{x})}{\alpha}(1 + c) \leq 1 \iff c < \frac{1}{Q(\hat{x})M} - 1.$$

$\square$

# B  ADDITIONAL RESULTS FOR BERNOULLI DISTRIBUTIONS

We present additional results and plots that were left out of the main text due to the page limit. The additional results are consistent with the story discussed in the main text.

## B.1  DIRECT SAMPLING CALIBRATION PLOTS FOR OTHER LLMS

Figure 11 presents the calibration plots with various reasoning length constraints for $P_1$ for GPT-4.1-nano (left) and Qwen-2.5 72B (right). Overall, the models seem to be better calibrated for $p \in [0, 0.5]$ while showing a similar bias as Llama-3.1 70B (compare to Figure 5) across different reasoning lengths.

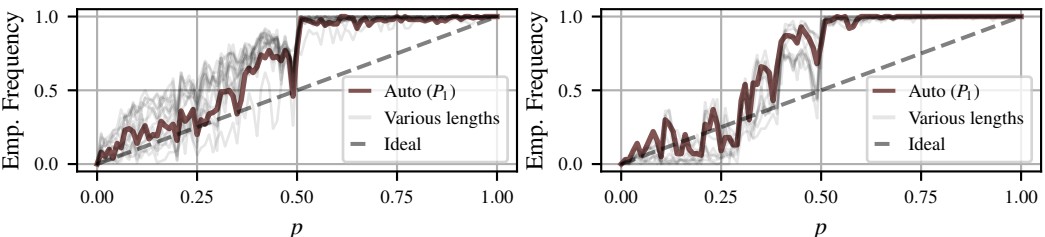

Figure 11: Calibration for various reasoning length constraints in direct sampling: GPT-4.1-nano (left) and Qwen-2.5 72B (right) for $P_1$.

Figure 12 shows the calibration plots of direct sampling for GPT-4.1-nano, Qwen-2.5 72B, and DeepSeekV3. The corresponding STVD scores are shown in Table 1. The corresponding VRS calibration plots are shown in Figure 13.

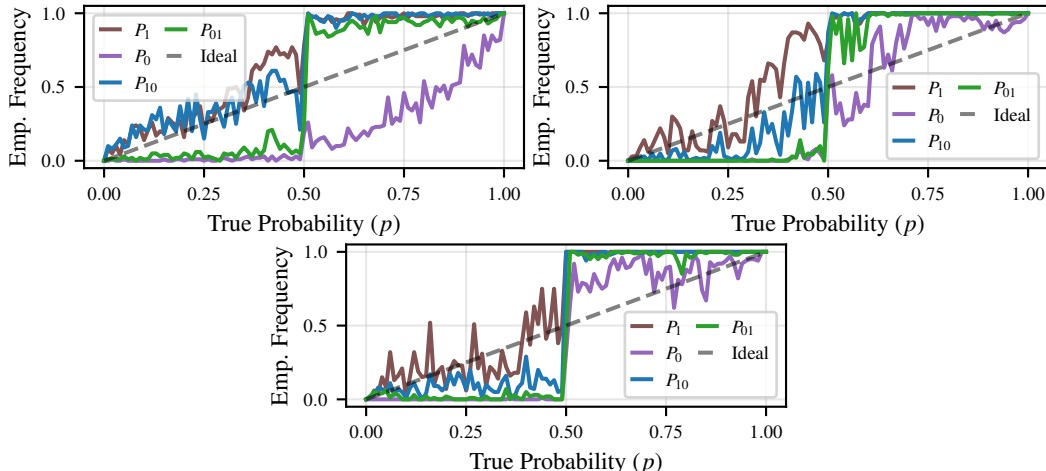

Figure 12: Calibration of direct sampling for GPT-4.1-nano (top left), Qwen-2.5 72B (top right), and DeepSeekV3 (bottom).

## B.2 VRS CALIBRATION PLOTS FOR OTHER LLMs

In Figure 13 we provide calibration plots of VRS for GPT-4.1-nano (top left), Qwen-2.5 72B (top right), and DeepSeekV3 (bottom). In Table 1 we provide the corresponding STVD. We find that the smaller GPT-4.1-nano performs worse than the other two larger models. However, the plots tell the same story as the ones in the main text.

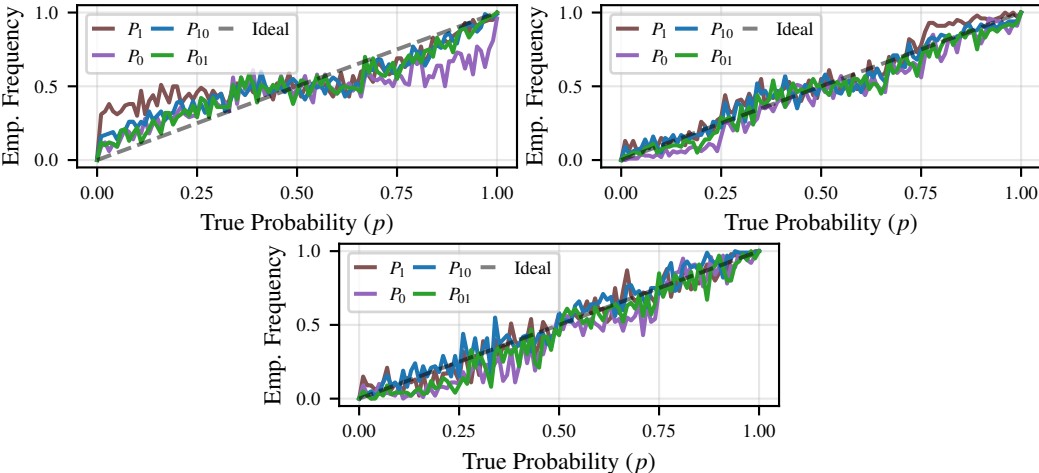

Figure 13: Calibration of VRS for GPT-4.1-nano (top left), Qwen-2.5 72B (top right), and DeepSeekV3 (bottom).

## B.3 VRS WITH CONSTANT $M$ INSTRUCTION

Figure 14 shows the calibration plot for VRS-M, which is a ablation of VRS by providing the model with the description on how $M$ is computed. The corresponding STVD scores can be found in Table 2.

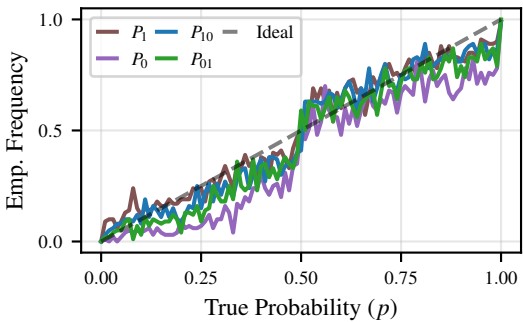

Figure 14: Calibration of VRS-M (Llama-3.1)

## B.4 FULL RESULTS FOR PROMPT ABLATIONS

We provide the full table corresponding to Table 2 that includes standard deviation over 5 independent trails for each method. The overall observation is the same as in Section 6.1. The low standard deviations indicate the observed effects are stable across runs.

Table 3: Ablation STVD with Standard Deviation($\downarrow$)

| Method | $P_1$ | $P_0$ | $P_{10}$ | $P_{01}$ | mean |
|---|---|---|---|---|---|
| Direct | $25.36 \pm 0.13$ | $24.79 \pm 0.49$ | $12.50 \pm 0.35$ | $16.59 \pm 0.45$ | $19.81 \pm 0.15$ |
| VRS | $5.73 \pm 0.29$ | $7.64 \pm 0.23$ | $5.36 \pm 0.10$ | $5.60 \pm 0.10$ | $6.08 \pm 0.12$ |
| VRS-simple | $15.53 \pm 0.30$ | $6.97 \pm 0.34$ | $13.99 \pm 0.16$ | $10.95 \pm 0.32$ | $11.86 \pm 0.13$ |
| VRS-simple-M | $11.19 \pm 0.36$ | $29.02 \pm 0.46$ | $14.08 \pm 0.68$ | $19.49 \pm 0.74$ | $18.45 \pm 0.35$ |
| VRS-M | $5.17 \pm 0.28$ | $11.40 \pm 0.42$ | $5.59 \pm 0.23$ | $7.28 \pm 0.22$ | $7.36 \pm 0.14$ |

## B.5 ABLATIONS FOR OTHER LLMS

We provide the full table corresponding to Table 2 that includes all four LLMs. The overall observation is the same as in Section 6.1. Adding the $M$-instruction leads to degradations of the sampling performance for both VRS-simple and VRS. Additionally, except for GPT-4.1-nano, for the three other LLMs, VRS-simple improve the performance on average, indicating that the VRS prompt can explain some of the improvement, but the effectiveness of the same prompt varies across LLMs. Therefore, the general improvement of VRS is likely to come from the algorithm itself, rather then prompt phrasing alone.

Table 4: Ablation STVD for all models($\downarrow$)

| Method | Llama-3.1 70B | | | | | GPT-4.1-nano | | | | | DeepSeekV3 | | | | | Qwen-2.5 72B | | | | |
|---|---|---|---|---|---|---|---|---|---|---|---|---|---|---|---|---|---|---|---|---|
| | $P_1$ | $P_0$ | $P_{10}$ | $P_{01}$ | mean | $P_1$ | $P_0$ | $P_{10}$ | $P_{01}$ | mean | $P_1$ | $P_0$ | $P_{10}$ | $P_{01}$ | mean | $P_1$ | $P_0$ | $P_{10}$ | $P_{01}$ | mean |
| Direct | 25.36 | 24.79 | 12.50 | 16.59 | 19.81 | 17.87 | 30.23 | 16.63 | 19.24 | 21.00 | 17.76 | 19.39 | 20.78 | 23.26 | 20.30 | 20.73 | 18.72 | 19.00 | 22.64 | 20.27 |
| VRS | 5.73 | 7.64 | 5.36 | 5.60 | 6.08 | 12.96 | 13.06 | 9.50 | 8.46 | 11.00 | 5.34 | 9.06 | 5.29 | 6.94 | 6.66 | 5.93 | 6.35 | 4.49 | 5.12 | 5.47 |
| VRS-simple | 15.53 | 6.97 | 13.99 | 10.95 | 11.86 | 36.83 | 22.05 | 25.56 | 22.29 | 26.68 | 8.23 | 20.25 | 11.19 | 15.59 | 13.82 | 13.55 | 8.21 | 10.69 | 8.69 | 10.28 |
| VRS-simple-M | 11.19 | 29.02 | 14.08 | 19.49 | 18.45 | 29.08 | 20.68 | 29.97 | 28.23 | 26.99 | 18.25 | 30.43 | 20.82 | 28.49 | 24.50 | 14.43 | 9.48 | 12.41 | 10.13 | 11.61 |
| VRS-M | 5.17 | 11.40 | 5.59 | 7.28 | 7.36 | 10.3 | 12.29 | 12.39 | 12.77 | 11.94 | 8.79 | 14.33 | 8.73 | 11.26 | 10.78 | 8.92 | 9.66 | 8.78 | 9.55 | 9.23 |

## C ADDITIONAL RESULTS FOR BINOMIAL DISTRIBUTIONS

In this section, we extend our results in to Binomial distributions. The binomial distribution $P$ has parameters $p, n$ and the probability mass function is given by

$$P(k) = \binom{n}{k} p^k (1-p)^{n-k}.$$

For two Binomial distributions $P$ and $Q$ with parameters $n, p$ and $n, q$, respectively, we have

$$M := \max_{k \in \{0, \dots, n\}} \left\{ \left( \frac{p}{q} \right)^k \left( \frac{1-p}{1-q} \right)^{n-k} \right\}. \tag{10}$$

In the following Appendix C.1, we show that both Proposition 1 and Proposition 2 generalize to Binomial distributions (as well as other distributions with discrete state spaces, same distributional form of $P$ and $Q$, and a single outcome that maximizes the acceptance ratio) under the assumption that the LLM arrives at the right decision if the acceptance ratio is 1 (which we observe to be true for the Bernoulli case). Additionally, the bound presented in Corollary 1 also generalizes under the assumption that direct sampling yields a distribution within TV distance $c$ of the target. In Appendix C.2, we verify the theoretical statements empirically and we observe that VRS can generate samples with smaller TV error than direct sampling under several settings.

### C.1 EXTENDING THE THEORETICAL ANALYSIS

We will establish a worst-case bound in terms of the TV distance between two Binomial distributions $P$ and $Q$ for the case that the acceptance probability is biased for every $k$.

**Proposition 3.** *Let $P(k), Q(k)$ be Binomial distributions with parameters $n, p$ and $n, q$, respectively, where $P$ is the target distribution that we want to sample from with rejection sampling and $Q(k)$ is the proposal distribution. Further, let $\tilde{P}(k)$ denote the Binomial distribution resulting from a biased accept/reject step where we assume that the acceptance probability $\tilde{A}(k)$ is biased as $\tilde{A}(k) = A(k) + e(k)$ where $|e(k)| \leq c \in \mathbb{R}$. Then,*

$$D_{\mathrm{TV}}(\tilde{P}, P) \leq \frac{Mc}{1 - Mc},$$

*where $M$ is defined as in Equation (10).*

*Proof.* Let $B := \{0, \dots, n\}$. Note that $\alpha = \sum_{k \in B} Q(k)A(k) = \frac{1}{M}$. Assuming a biased acceptance probability $\tilde{A}(x)$, we can split the resulting acceptance rate into

$$\tilde{\alpha} = \sum_{k \in B} Q(k)(A(k) + e(k)) = \underbrace{\sum_{k \in B} Q(k)A(k)}_{=\alpha} + \underbrace{\sum_{k \in B} Q(k)e(k)}_{=:\delta},$$

where $\alpha$ corresponds to the unbiased acceptance rate and $\delta$ denotes the deviation from it. Note that

$$|\delta| = \sum_{k \in B} Q(k)|e(k)| \leq \sum_{k \in B} Q(k)c = c,$$

and, therefore, $\tilde{\alpha} = \alpha + \delta \geq \alpha - c \geq 0$. We assume that $0 \leq \tilde{A}(k) \leq 1$. Let $\tilde{A}$ denote the acceptance event. We denote the resulting law of the accepted samples by $\tilde{P}$.

$$\mathbb{P}(K = k \mid \tilde{A}) = \frac{\mathbb{P}(K = k, \tilde{A})}{\mathbb{P}(\tilde{A})} = \frac{Q(k)\tilde{A}(k)}{\tilde{\alpha}} =: \tilde{P}(k).$$

We can now upper-bound a term in the TV distance as

$$|\tilde{P}(k) - P(k)| = \left| \frac{Q(k)\tilde{A}(k)}{\tilde{\alpha}} - \frac{Q(k)A(k)}{\alpha} \right| \leq Q(k)\frac{c}{\alpha - c}\left(1 + \frac{A(k)}{\alpha}\right),$$

which follows as in Proposition 1. For the full TV distance, we get

$$D_{\text{TV}}(\tilde{P}, P) = \frac{1}{2} \sum_{k \in B} \frac{c}{\alpha - c} Q(k) \left( 1 + \frac{A(k)}{\alpha} \right)$$

$$= \frac{1}{2} \frac{c}{\alpha - c} \left( \sum_{k \in B} Q(k) + \frac{1}{\alpha} \sum_{k \in B} Q(k) A(k) \right)$$

$$= \frac{c}{\alpha - c}$$

$$= \frac{Mc}{1 - Mc}.$$

$\square$

Note that we can extend Proposition 3 to an arbitrary distribution with finite state space.

In the following, we will assume that if $A(k^*) = 1$ there is no bias, i.e., no error ($A(k^*) = 1 \Rightarrow e(k^*) = 0$). Additionally, we assume that there exists only one $k^*$ which achieves $A(k^*) = 1$. For all other $k \in B \backslash \{k^*\}$, we have $|e(k)| \leq c$.

**Proposition 4.** *Let $P(k), Q(k)$ be Binomial distributions with parameters $n, p$ and $n, q$, respectively, where $P(k)$ is the target distribution that we want to sample from with rejection sampling and $Q(k)$ is the proposal distribution. Further, let $\tilde{P}(k)$ denote the Bernoulli distribution resulting from a biased accept/reject step where we assume that the acceptance probability $\tilde{A}(k)$ is biased with an additive error $e(k)$ where $|e(k)| \leq c \in \mathbb{R}$ as*

$$\tilde{A}(k) = \begin{cases} A(k) + e(k) & \text{if } A(k) < 1 \\ A(k) & \text{if } A(k) = 1 \end{cases},$$

*where $|e(x)| \leq c \in \mathbb{R}$. Further, we assume that there exists only one $k^*$, such that $A(k^*) = 1$. Then,*

$$D_{\text{TV}}(\tilde{P}, P) \leq \frac{Mc\bar{q}_{k^*}}{1 - Mc\bar{q}_{k^*}}$$

*where $M$ is defined as in Equation (10) and $\bar{q}_{k^*} = 1 - Q(k^*)$.*

*Proof.* Let $k^*$ be chosen such that $A(k^*) = 1 \Rightarrow e(k^*) = 0$. The resulting acceptance rate $\tilde{\alpha}$ can be states as follows

$$\tilde{\alpha} = \sum_{k \in B} Q(k)(A(k) + e(k)) = \underbrace{\sum_{k \in B} Q(k) A(k)}_{= \alpha} + \underbrace{\sum_{k \in B \backslash \{k^*\}} Q(k) e(k)}_{=: \delta}.$$

For any $k$, we have

$$|\tilde{P}(k) - P(k)| = Q(k) \left| \frac{e(k)\alpha - A(k)\delta}{\alpha\tilde{\alpha}} \right|$$

For $k^*$, we have $e(k^*) = 0, A(k^*) = 1$ and, therefore,

$$|\tilde{P}(k^*) - P(k^*)| = Q(k^*) \left| \frac{e(k^*)\alpha - A(k^*)\delta}{\alpha\tilde{\alpha}} \right| = Q(k^*) \frac{|\delta|}{\alpha\tilde{\alpha}}$$

For $k \in B \backslash \{k^*\}$ we get

$$\sum_{k \in B \backslash \{k^*\}} |\tilde{P}(k) - P(k)| = \sum_{k \in B \backslash \{k^*\}} Q(k) \left| \frac{e(k)\alpha - A(k)\delta}{\alpha\tilde{\alpha}} \right|$$

$$\leq \sum_{k \in B \backslash \{k^*\}} Q(k) \frac{|e(k)|\alpha + A(k)|\delta|}{\alpha\tilde{\alpha}}$$

$$= \frac{1}{\alpha\tilde{\alpha}} \left( \alpha \sum_{k \in B \backslash \{k^*\}} Q(k)|e(k)| + |\delta| \sum_{k \in B \backslash \{k^*\}} Q(k) A(k) \right)$$

Summing over all $k \in B$, we get

$$
\begin{aligned}
2D_{\text{TV}}(\tilde{P}, P) &= \sum_{k \in B} |\tilde{P}(k) - P(k)| \\
&= |\tilde{P}(k^*) - P(k^*)| + \sum_{k \in B \setminus \{k^*\}} |\tilde{P}(k) - P(k)| \\
&\leq Q(k^*) \frac{|\delta|}{\alpha\tilde{\alpha}} + \frac{1}{\alpha\tilde{\alpha}} \left( \alpha \sum_{k \in B \setminus \{k^*\}} Q(k)|e(k)| + |\delta| \sum_{k \in B \setminus \{k^*\}} Q(k)A(k) \right) \\
&= \frac{1}{\alpha\tilde{\alpha}} \left( \alpha \sum_{k \in B \setminus \{k^*\}} Q(k)|e(k)| + |\delta| \left( Q(k^*) + \sum_{k \in B \setminus \{k^*\}} Q(k)A(k) \right) \right) \\
&= \frac{1}{\alpha\tilde{\alpha}} \left( \alpha \sum_{k \in B \setminus \{k^*\}} Q(k)|e(k)| + |\delta|\alpha \right) \\
&= \frac{2}{\alpha\tilde{\alpha}} \alpha|\delta| \\
&\leq \frac{2c\bar{q}_{k^*}}{\tilde{\alpha}},
\end{aligned}
$$

using

$$
\sum_{k \in B \setminus \{k^*\}} Q(k)A(k) = \alpha - Q(k^*),
$$

and the upper-bound on $|\delta|$

$$
|\delta| = \sum_{k \in B \setminus \{k^*\}} Q(k)|e(k)| \leq c \sum_{k \in B \setminus \{k^*\}} Q(k) = c(1 - Q(k^*)) =: c\bar{q}_{k^*}.
$$

We have

$$
\tilde{\alpha} \geq \alpha - |\delta| \geq \alpha - c\bar{q}_{k^*}.
$$

We can rewrite this bound in terms of $M$ as follows.

$$
D_{\text{TV}}(\tilde{P}, P) \leq \frac{c\bar{q}_{k^*}}{\tilde{\alpha}} = \frac{c\bar{q}_{k^*}}{\alpha - c\bar{q}_{k^*}} = \frac{Mc\bar{q}_{k^*}}{1 - Mc\bar{q}_{k^*}}
$$

$\square$

Comparing the bounds in this section to the bounds in the main text, we note that they are similar.

**Remark 1** (Comparing the bounds to the main result.)**.** *While the bound in Proposition 3 is the same as Proposition 1, the bound in Proposition 4 is a generalization of the bound in Proposition 2 by replacing $Q(\hat{x}) = 1 - Q(x^*)$ by $1 - Q(k^*)$ (note that there are generally more than two k).*

With the derivations in this section, we can also generalize the bounds to other distributions.

**Remark 2.** *(Generalizing the bounds.) Note that we can extend Proposition 3 to other distributions with discrete state-space. Also, we can extend Proposition 4 to other distributions with discrete state space, under the assumption that $P$ and $Q$ are the same distribution and that there is only a single $k^*$ that achieves the condition $A(k^*) = 1$. The latter condition is a worst case condition, i.e., if there is more than one $k^*$ with $A(k^*) = 1$, the bound presented in Corollary 2 is looser.*

In the following corollary we compare the worst case bounds derived in Proposition 4 to a general error in TV distance of $c \in \mathbb{R}$.

**Corollary 2.** *Let $P(k), Q(k)$ be Binomial distributions with parameters $n, p$ and $n, q$, respectively, where $P(k)$ is the target distribution that we want to sample from with rejection sampling and $Q(k)$ is the proposal distribution. Further, let $\tilde{P}(k)$ denote the Binomial distribution resulting from a*

*biased accept/reject step where we assume that the acceptance probability $\tilde{A}(k)$ is biased with an additive error $e(k)$, where $|e(k)| \leq c \in \mathbb{R}$, if $A(k) < 1$, see Proposition 4. Additionally, let $\bar{P}(k)$ be the Binomial distribution for which we assume*

$$D_{\mathrm{TV}}(\bar{P}, P) \leq c.$$

*Then, the worst-case total variation error of half-biased rejection sampling is smaller than that of direct sampling if and only if*

$$D_{\mathrm{TV}}(\tilde{P}, P) < D_{\mathrm{TV}}(\bar{P}, P) \iff \frac{\bar{q}_{k^*}}{\alpha}(1 + c) \leq 1 \iff c < \frac{1}{\bar{q}_{k^*} M} - 1.$$

*where $M$ is defined as in Equation (10) and $\bar{q}_{k^*} = 1 - Q(k^*)$.*

*Proof.* Let

$$D_{\mathrm{TV}}(\bar{P}, P) = |e(x)| \leq c, \tag{11}$$

and

$$D_{\mathrm{TV}}(\tilde{P}, P) \leq \frac{Mc\bar{q}_{k^*}}{1 - Mc\bar{q}_{k^*}}, \tag{12}$$

following the bound in Proposition 4. Then,

$$\frac{Mc\bar{q}_{k^*}}{1 - Mc\bar{q}_{k^*}} < c \iff \frac{\bar{q}_{k^*}}{\alpha}(1 + c) \leq 1 \iff c < \frac{1}{\bar{q}_{k^*} M} - 1.$$

$\square$

In Corollary 2, we assume $D_{\mathrm{TV}}(\bar{P}, P) \leq c$ instead of an additive error on the parameter of the Bernoulli distribution as in Corollary 1.

**Remark 3.** *(Assumptions in Corollary 2.) In Corollary 2 we assume that the distribution $\bar{P}$ (eventually sampled from the LLM) is biased by $D_{\mathrm{TV}}(\bar{P}, P) \leq c$ which is in contrast to Corollary 1 where we assume an additive error on the parameter $\bar{p}$, i.e., $\bar{p} = p + e$, $|e| \leq c$. This is due to the fact that in the Bernoulli case (Corollary 1), we have a clear picture on the functional form of the error (an additive shift on the parameter). However, in the Binomial case (and in the case of other discrete distributions), we do not have an idea on how the error comes about, instead we assume an error budget of $c \in \mathbb{R}^+$ measured in TV distance. For more informed bounds, future work might investigate the structure of samples, given by LLMs, for other distributions.*

## C.2 EMPIRICAL EVALUATION OF VRS ON BINOMIAL DISTRIBUTIONS

We evaluated VRS v.s. direct sampling on $Q$ being $\mathrm{Binomial}(n, 0.5)$ and $P$ being $\mathrm{Binomial}(n, p)$ for $n \in \{1, 2, 3, 4, 5\}$, across 11 values of $p \in \{0.0, 0.1, ..., 1.0\}$ (unlike in the main text which is across 101 values of $p$). Resulting STVD $\downarrow$ (summed over all $p$ values) for Llama-3.1 70B is showed in Table 5. We can see that with larger $n$, i.e., the distribution being more complex, the STVD for both direct sampling and VRS is getting larger. Nevertheless, the VRS still results in much smaller STVD than direct sampling.

Table 5: STVD ($\downarrow$) for Binomial distributions.

| Method | $n = 1$ | $n = 2$ | $n = 3$ | $n = 4$ | $n = 5$ |
|--------|---------|---------|---------|---------|---------|
| Direct | 1.32 | 2.50 | 3.89 | 3.78 | 4.08 |
| VRS | 0.52 | 1.19 | 2.23 | 2.46 | 2.74 |

# D EXPERIMENT SETUP DETAILS

In this section, we provide the pseudocode algorithms for VRS in the setting of Bernoulli, and the details for the computational resources used for our experiments.

## D.1 PSEUDOCODE FOR VRS WITH BERNOULLI

---
**Algorithm 1** VRS for Bernoulli

---
Given: language descriptions for the target $P(x; p)$, language descriptions for the proposal $Q(x; 0.5)$, number of samples $N$;

$samples$ = [];
**for** $n = 1, \cdots, N$ **do**
   **repeat**
      $s \sim \text{Bern}(0.5)$;             // Python Sampler
      $\text{resp} = \text{LLM}(P, Q, s, \text{template})$;     // LLM API call
   **until** $\text{resp} = \text{T}$          // T *for 'Accept',* F *for 'Reject'*;
   $samples$.append($s$);
**end**
**return** $samples$

---

## D.2 COMPUTATIONAL RESOURCES

We host open-source models (e.g., Llama-3.1 70B and Qwen-2.5 72B) using the vLLM [9] framework on 4 Nvidia A100 GPUs or 4 Nvidia H100 GPUs. Generating 100 samples from the LLMs takes approximately 25 seconds in our setup.

**Licenses** For the open-source models, we use Llama-3.1 (LLAMA 3.1 COMMUNITY LICENSE AGREEMENT), DeepSeekV3 (DEEPSEEK LICENSE AGREEMENT), and Qwen-2.5 (Qwen LICENSE AGREEMENT). We buy the service from OpenAI to use GPT-4.1-nano.

## E  BROADER DISCUSSIONS

We do recognize the idea of using natural language to define distributions and using LLMs as samplers for such distributions is very much at its early stage, and it is not fully recognized by many at the moment. However, as LLMs are playing an increasingly larger role in computing, their ability to generate faithful samples has to be studied and improved, which is closely tied to topics like fairness and safety. We are not hoping to solve this problem entirely in a single work, instead, we want to use the simplest distribution, i.e., the Bernoulli, to raise the awareness and thoroughly study this overlooked problem, and to investigate possible fixes.

During the project, we came across many interesting discussions, which we believe are worth sharing with the reader in this section. Some discussions are formatted in Q&A below.

### E.1  WHY NOT ASK THE LLM TO CALL AN EXTERNAL SAMPLER?

We fully agree that when an external tool, e.g., Python-based sampler, is available and callable, it can be used to generate unbiased samples efficiently. However, our work is not positioned as a replacement for such tool-assisted approaches. Instead, we focus on a different and complementary question:

*"Can large language models, operating solely in natural language, simulate stochastic processes faithfully without access to external tools or code?"*

This question arises from realistic and increasingly common LLM deployment scenarios where: LLMs act as autonomous *agents* expected to make decisions involving chance (e.g., tie-breaking, randomized planning), interfaces are purely natural language, with no tool execution available or permitted, even when tools are available, their invocation may compromise interpretability, modularity, or security (e.g., sandboxed educational or fairness-sensitive settings).

In such settings, we are left with the LLM itself as the only accessible computational mechanism. The goal of VRS is to explore whether LLMs can simulate stochasticity internally, using only structured prompting. This is not about generating perfect randomness, but about understanding and improving the LLM's native stochastic behavior, an ability that is underexplored but increasingly relevant as LLMs are deployed as autonomous agents and decision-makers.

**More broadly**, VRS is not just a sampling method, *it is a case study in how to build and analyze algorithmic prompts in a principled way.* Rather than relying on heuristic prompt engineering, we derive a prompt-based implementation of rejection sampling and provide formal theoretical guarantees for its behavior under model bias. This methodology, i.e., analyzing prompts through the lens of classical algorithms and error bounds, offers a new paradigm for prompt design that bridges empirical performance with formal analysis.

**The analogy here is research on LLMs' math capabilities**: although it's trivial to solve math problems by calling a calculator, we still study whether LLMs can reason through equations in language, because it tells us something fundamental about their internal representations and limitations. In the same spirit, VRS asks whether LLMs can simulate randomness themselves, not by outsourcing it, but by verbalizing and executing probabilistic logic in language.

### E.2  HOW PRACTICAL IS IT TO USE LLMS AS SAMPLERS?

This work represents an early step toward enabling and understanding more advanced verbalized probabilistic algorithms. The setup in this paper, focused on Bernoulli distributions, is intentionally simple, not because the problem is trivial, but because it offers a concrete foundation to study a deep and emerging capability: *probabilistic reasoning in natural language*.

We fully agree that in classical settings, sampling should rely on tools, which offer well-defined guarantees. However, our motivation arises from realistic and increasingly common LLM deployment scenarios where: LLMs act as autonomous *agents* expected to make decisions involving chance (e.g., tie-breaking, randomized planning), interfaces are purely natural language, with no tool execution available or permitted, even when tools are available, their invocation may compromise interpretability, modularity, or security (e.g., sandboxed educational or fairness-sensitive settings).

While invoking an external tool is technically sound, relying on tool use as a universal solution may not be realistic or sufficient. Many LLM-based systems already operate in tool-free settings, and users (often unknowingly) trust the LLM's verbal reasoning to simulate stochasticity. We view VRS not as a replacement for principled samplers, but as a practical, language-native safeguard that significantly reduces sampling error in such environments.

In the long term, this raises several foundational questions: How can we understand and control stochastic behavior in LLMs through reasoning? What are the algorithmic abstractions that can be embedded within language? How robust are these probabilistic reasoning? These are open and important challenges. LLMs currently do not offer distributional guarantees. But if we want to reason about and improve their probabilistic reasoning capabilities, we must begin somewhere, and this work aims to provide that conceptual and empirical example.

Finally, while VRS currently assumes access to an explicit target distribution $P(x)$, a compelling future direction is to extend this framework to implicitly defined distributions, where $P(x)$ is only described semantically or via examples, rather than analytically. In such cases, tool use may no longer help, as there is no closed-form function to evaluate. Interestingly, our findings in Section 4.1 show that LLMs are often better at recognizing whether a sample fits a distribution than at generating it. This discriminator-like ability could inspire new verbalized sampling paradigms, perhaps analogous to adversarial models like GANs, where judgment about sample quality is used to refine generative behavior.

In short, we agree that faithful sampling from LLMs is a difficult and unresolved problem, but it is precisely because it is difficult, and increasingly relevant, that we believe it deserves attention now.

### E.3 DOES VRS INFLUENCE OR ALIGN THE LLM'S INTERNAL LOGITS?

This question gets to the heart of why we believe VRS is both interesting and distinct from other approaches.

**VRS does not modify or align logits.** Unlike direct sampling, where the LLM is prompted to output "0" or "1" and the resulting logits directly correlate with the sample distribution, VRS prompts the model to sample a decision to accept (T) or reject (F) the proposed sample. Thus, VRS operates over a different output space and does not influence or depend on the logits used in direct sampling.

While VRS does not try to align the model's internal probabilities, it does yield samples that better match the target distribution. As shown in Section 6.2, this is not due to logit manipulation but to the algorithmic structure imposed by the prompt. This distinguishes VRS from tool-calling (which delegates randomness to external code) and from methods requiring internal model access.

**VRS intentionally accepts biased logits and works around them.** In contrast to methods that modify LLM behavior by adjusting weights (via fine-tuning) or prompts (via prompt engineering), VRS embraces the fact that the logits are biased and uses a probabilistic mechanism (executed by LLMs) to correct for it, without needing access to or control over the internal distributions. This is conceptually aligned with classical sampling theory: for decades, rejection sampling and similar methods have been used to generate unbiased samples from biased sources. VRS brings this idea to the language interface of LLMs.

### E.4 IS VRS COMPUTATIONALLY INTENSIVE?

VRS incurs higher computational cost than direct sampling, and this is both expected and meaningful. The key point, however, is that *VRS is an algorithm that operates in a fundamentally different space: the informal, natural language domain*.

In classical settings, sampling algorithms like rejection sampling or MCMC are implemented in formal programming environments (e.g., Python). These are efficient, but they also assume the user has already formalized the problem, encoded it in code, and specified exact parameters (e.g., manually calculated the acceptance probability). In contrast, VRS addresses a very different use case: the user specifies the problem informally in natural language, and the computation itself happens within the language space.

This shift, what we might call **verbalized computing**, has important implications. While inference in this space may be more computationally expensive, it is also more accessible. A user can invoke VRS by simply describing a desired distribution in plain text. There is no need to write or call code, craft sampling functions, or define rejection logic programmatically. This convenience is not free, but it lowers the barrier of entry for users who otherwise would not engage with formal stochastic computation. Viewed this way, **the "cost" of VRS is offset by the elimination of the cost of formalization, which is often unacknowledged in computational models, yet a dominant factor in practice.**

Moreover, *we believe this reframes how we think about computational complexity in the LLM era*. Traditional complexity theory does not account for the cost of formalization, and directly starts with an already formalized problem to analyze its complexity. In LLM-based systems, where both problem specification and computation occur in natural language, the relevant complexity includes the effort saved by not formalizing the task, and that's where natural-language-based algorithms like VRS shines.

We also want to clarify that the actual sampling overhead of VRS is bounded and modest in the Bernoulli case. Since we use a symmetric proposal distribution (i.e., $q = 0.5$), the worst-case acceptance probability is $0.5$, meaning we expect to draw twice as many proposals as needed samples in the worst case. This remains in the same complexity class: generating $n$ accepted samples via VRS still takes $\mathcal{O}(n)$ calls to the model.

### E.5   DOES USING A PROGRAMMATIC SAMPLER FOR THE PROPOSAL $Q$ WEAKEN THE RESULT?

In our implementation, the proposed sample $s \sim Q$ (where $Q = \text{Bern}(0.5)$) is generated programmatically. This can be done using standard libraries (e.g., Python's random module) or deterministically, for example by submitting half the prompts with $s = 1$ and the other half with $s = 0$. Nevertheless, we believe it does not weaken the results for the following reasons:

**The LLM's stochastic behavior is still central.**   A crucial step in our method is whether the LLM can reliably carry out the accept/reject decision in a probabilistic way, purely through reasoning over language. This is precisely where LLMs have struggled in direct sampling, and where VRS shows a surprising improvement. The fact that the input $s$ is sampled externally does not diminish this core finding.

**Programmatic randomness is standard in computational sampling.**   Virtually all stochastic processes in simulations or machine learning, whether it's sampling from a Gaussian, Bernoulli, or any complex distribution, ultimately rely on deterministic procedures to generate pseudo-randomness. For example, diffusion models begin with noise sampled from a programmatic Gaussian, which is then transformed into structured outputs (e.g., images).

**VRS mirrors this classical setup.**   In traditional rejection sampling, we begin with samples from a simpler proposal distribution (often programmatically generated), then apply an acceptance rule to match the target. VRS follows this paradigm: $s \sim Q$ comes from a simple source, while the LLM plays the key role of evaluating and filtering these proposals to better approximate $P$.

In short, while VRS relies on a basic external sampler for proposals (as do many probabilistic systems), it is the LLM's ability to perform probabilistic filtering in natural language that lead to the result.

### E.6   IS VRS A FEW-SHOT PROMPTING METHOD?

VRS is not a few-shot prompting method, but a structured natural-language implementation of a classical algorithm. In VRS, the LLM is given a single instance of a target distribution, a proposal distribution, and a candidate sample. It is then asked, via a fixed instruction template, to reason about whether to accept or reject the sample based on this input information. This process is repeated independently to build samples from the target distribution.

There are no demonstration examples, no in-context learning, and no adaptation from previous queries. Instead, the LLM is executing a natural-language instruction (accept/reject logic) conditioned on input

values, resembling algorithmic reasoning more than imitation of the inputs (also, there is nothing to imitate with). Each invocation of VRS is stateless, self-contained, and purely instructional.

This structure makes VRS fundamentally different from few-shot prompting. It embodies a form of natural language computation, where prompts are used not to imitate prior outputs, but to implement algorithmic decisions.

# F EXAMPLE SAMPLING LOGS

## F.1 DIRECT SAMPLING (LLAMA-3.1, COT LENGTH 'AUTO', $P_1(x = 1) = 0.75$)

---

**Sampling History: Direct Sampling**

——————— **Sampling Prompt** ———————
You are a sampler. You will come up with a sample based on the descriptions below.

**\*\* Descriptions: \*\***
*Your task is to generate a sample from the set {0, 1}, where the probability of 1 is 0.75.*

Please give your output strictly in the following format:

```
Explanations:
[Your step-by-step analyses and results; You DO NOT have access to a computer or a random number generator]

Output:
[Your output MUST be an integer in {0, 1}]
```

Please ONLY reply according to this format, don't give me any other words.

——————— **Sampling Output (1 out of 100 queries)** ———————
Explanations:
Since the probability of 1 is 0.75, we can use a method like "quartering" to make a decision. Flip a fair coin three times. If all three flips are heads, the result is 0. If any of the flips are tails, the result is 1.

Assuming the flips are: tails, heads, tails, which is a valid outcome.

Output:
1

——————— **Sampling Output (1 out of 100 queries)** ———————
Explanations:
To generate a sample from the set {0, 1} with a probability of 1 being 0.75, we can use a simple manual method. Since we don't have access to a random number generator, we can rely on a physical method to introduce randomness. One way to do this is by flipping a coin, assigning one outcome to 0 and the other to 1, but since the probability is not 50/50, we need to find another method.
Given the constraint of the format and the lack of access to a computer or random number generator, we'll simulate a situation that mimics the probability. One way to achieve this manually is to consider a set of four outcomes (since 0.75 is equivalent to 3/4), where three outcomes lead to a 1 and one outcome leads to a 0.
Let's assign these outcomes to the numbers 1-4: 1, 2, and 3 for 1; 4 for 0.

Output:
1

**Many outputs for the rest i.i.d. queries ...**

——————— **Sampling Output (1 out of 100 queries)** ———————
Explanations:
Given the probability of 1 is 0.75, I will simulate this by dividing the interval [0, 1) into two parts: [0, 0.75) for 1 and [0.75, 1) for 0. Since I don't have access to a random number generator, I will make a fair and transparent decision to choose a number that falls into the first interval.

Output:
1

---

## F.2 VRS (LLAMA-3.1, COT LENGTH 'AUTO', $P_1(x = 1) = 0.75$)

---

**Sampling History: Verbalized Rejection Sampling**

——————— **Sampling Prompt (the proposed sample being 1)** ———————
You are a rejection sampler. Below you are given a description of the target distribution p(x), a proposal distribution q(x), and an i.i.d. sample from q(x). You need to decide whether or not to accept the sample.

**\*\* Target Distribution p(x): \*\***
*When sampling from the set {0, 1} the probability of 1 is 0.75.*

**\*\* Proposal Distribution q(x): \*\***
*A Bernoulli distribution with probability of having 1 in the set of {0,1} being 0.5*

**\*\* Sample from q(x): \*\***
1

Please give your output strictly in the following format:

'''
Explanations:
[Your step-by-step analyses and results; You DO NOT have access to a computer or a random number generator]

Output:
[Your output MUST be a letter in {T, F}]
'''

Please ONLY reply according to this format, don't give me any other words.

——————— **Sampling Output when the proposed sample is 1 (1 out of many queries)** ———————
Explanations:
First, calculate the probability of the sample from the target distribution p(x) and the proposal distribution q(x). p(1) = 0.75 and q(1) = 0.5. Next, calculate the acceptance probability. Since p(1)/q(1) = 0.75/0.5 = 1.5, which is greater than 1, we will always accept this sample.

Output:
T

——————— **Sampling Output when the proposed sample is 1 (1 out of many queries)** ———————
Explanations:
To determine whether to accept the sample, we need to calculate the probability of acceptance. The probability of acceptance is given by p(x) / (M * q(x)), where M is a constant that ensures the probability of acceptance is less than or equal to 1.
First, we need to find M. The maximum value of p(x) is 0.75, and the maximum value of q(x) is 0.5. Therefore, M = max(p(x)) / min(q(x)) = 0.75 / 0.5 = 1.5.
Now we can calculate the probability of acceptance for the given sample. p(1) = 0.75 and q(1) = 0.5. Therefore, the probability of acceptance = p(1) / (M * q(1)) = 0.75 / (1.5 * 0.5) = 1.
Since the probability of acceptance is 1, we accept the sample.

Output:
T

**Many outputs for the rest i.i.d. queries ...**

——————— **Sampling Output when the proposed sample is 0 (1 out of many queries)** ———————
Explanations:
To decide whether to accept the sample from q(x), we need to calculate the probability of acceptance using the formula for rejection sampling. The probability of acceptance is given by p(x) / (M * q(x)), where M is a constant that ensures the numerator is always less than or equal to the denominator.

First, we need to calculate the value of M. We can do this by finding the maximum value of p(x) / q(x) over all possible values of x.

For x = 0, p(0) = 0.25 and q(0) = 0.5, so p(0) / q(0) = 0.5.
For x = 1, p(1) = 0.75 and q(1) = 0.5, so p(1) / q(1) = 1.5.

Therefore, the maximum value of p(x) / q(x) is 1.5, so we can set M = 1.5.
Now, we can calculate the probability of acceptance for the sample x = 0:
p(0) / (M * q(0)) = 0.25 / (1.5 * 0.5) = 0.25 / 0.75 = 0.3333.
Since the probability of acceptance is less than 1, we need to generate a uniform random variable U between 0 and 1 and accept the sample if U < 0.3333. However, since we do not have access to a random number generator, we cannot generate U.

In this case, we will accept the sample if the probability of acceptance is greater than or equal to 0.5 (a rough estimate). Since 0.3333 < 0.5, we reject the sample.

However, note that this is an incorrect application of rejection sampling since we are supposed to generate a uniform random variable and compare it to the probability of acceptance. Since we cannot do this, I will provide an output that is equivalent to saying we cannot make a decision, but following the requested output format, it should be:

Output:
F

---

