# OpenReview forum: "What Can an LLM Flip if It Fails to Flip Coins?"
_ICLR.cc/2026/Conference — ICLR 2026 Conference Withdrawn Submission_

### Official Review · Reviewer_sQvG · 2025-10-17

**Soundness:** 2
**Presentation:** 2
**Contribution:** 1
**Rating:** 2
**Confidence:** 4

**Summary:**

This paper introduces a verbalized rejection sampling method in which LLMs are prompted to generate random samples from a target distribution. This method contrasts with verbalized direct sampling, where LLMs are directly prompted to produce random samples from the target distribution. In verbalized rejection sampling, the LLM is prompted to propose a sample from a simpler distribution Q, and then to decide whether to accept or reject the proposal when both the proposal distribution Q and the target distribution P are provided in context. Importantly, the entire process is carried out within a single prompt: there is no explicit second stage of rejection, and the rejection sampling procedure is completed in one step. The authors demonstrate that verbalized rejection sampling mitigates some of the biases previously observed in verbalized direct sampling. They also provide theoretical arguments for why this method performs better, highlighting that explicitly prompting LLMs to act as rejection samplers is the key contributing factor.

**Strengths:**

The proposed verbalized rejection sampling method is interesting. Investigating the ability of LLMs to generate random samples is an important topic, and this paper makes a useful contribution to that line of research.

**Weaknesses:**

The contribution of the paper is rather limited. Verbalized rejection sampling is essentially a form of prompt engineering, where modifying the prompts allows the LLM to achieve better performance in random sample generation. The theoretical analysis suggests that improved performance arises because the LLM is explicitly prompted to consider acceptance and rejection in the context of both the proposal distribution Q and the target distribution P. An important control experiment, in my view, would be to prompt LLMs to sample directly from P (as in verbalized direct sampling), while also asking them to consider whether to accept or reject the sample --- but without reference to Q. This condition could be regarded as verbalized rejection sampling without Q, or alternatively, as verbalized direct sampling augmented with a superficial requirement to evaluate whether the sample should be retained. Including such an intermediate condition would potentially strengthen the paper’s overall contribution.

**Questions:**

Do the authors also provide a theoretical analysis of verbalized direct sampling? In particular, what explains the systematic biases observed in verbalized direct sampling across different LLMs?

---

> ### Author Response · Authors · 2025-11-28
>
> We thank the reviewer for their time and assessment and for raising the interesting questions.
>
> > 1. "Just Prompt Engineering" vs. Algorithmic Implementation
>
> We respectfully distinguish VRS from standard "prompt engineering." While prompt engineering often involves heuristic search for optimal phrasing (e.g., "Think step by step"), VRS is the natural language implementation of a specific statistical algorithm.The prompt in VRS is not merely a "nudge"; it serves as the code that executes the Rejection Sampling algorithm (defining Target $P$, Proposal $Q$, and the Acceptance Logic).Our theoretical analysis (Section 6.2 and Appendix A) and ablation studies (Section 6.1) demonstrate that the performance gains come from the algorithmic guarantees of Rejection Sampling (filtering a biased source via a ratio), not simply from "better wording." As shown in Figure 10, the error is bounded by the algorithm's parameters ($M$ and $c$), a property that standard prompt engineering does not provide.
>
> > 2. The Suggested Control Experiment (Removing $Q$)
>
> Theoretically, if one samples directly from $P$ (Direct Sampling) and treats it as the proposal, then implicitly $Q = P$. In classical Rejection Sampling, if $Q = P$, the acceptance ratio $P(x)/(M \cdot Q(x))$ becomes $1/1 = 1$. The algorithm effectively collapses: every sample should be accepted. Therefore, the condition the reviewer describes, generating from $P$ and then asking "should I keep this?" without an alternative reference distribution $Q$, transforms the task from Rejection Sampling to Self-Correction (or Self-Verification), which then has no theoretical grounding on why some samples should be rejected.
>
> VRS works because it introduces an independent, simpler reference distribution ($Q$, e.g., a uniform coin flip) that acts as an anchor. The acceptance decision is a mathematical comparison between the target and this anchor. Prior work (e.g., Large Language Models Cannot Self-Correct Reasoning Yet) suggests that without external feedback or a structured alternative (like $Q$), LLMs struggle to "self-correct" their own intrinsic biases. If the model is biased in generating from $P$ (the "Knowledge-Sampling Gap"), it is likely biased in "verifying" $P$ without an external standard. While we view "Self-Correction" as a distinct mechanism from Rejection Sampling, we agree that discussing this distinction strengthens the paper. We will add a discussion clarifying that removing $Q$ reduces the method to self-correction, which lacks the statistical guarantees VRS provides.
>
> > 3. Theoretical Analysis of Direct Sampling Bias
>
> Our paper focuses on quantifying this bias (the "Knowledge-Sampling Gap" in Section 4.1) and correcting it (VRS), rather than deriving a new theory for why it exists. There are some hypotheses in recent literature that try to explain the phenomenon, but none of them provided theoretical analysis.
>
> The systematic bias (specifically the shift towards high-likelihood tokens or "1s" in our experiments) is best explained by the hypothesis that LLMs are trained (via Next Token Prediction) to maximize the likelihood of plausible answers. This objective aligns them with the mode of a distribution rather than its variance. VRS bypasses this by wrapping the sampling process within an algorithmic structure which provides a theoretical guarantee for the sampling bias.

---

### Official Review · Reviewer_rUw6 · 2025-10-31

**Soundness:** 2
**Presentation:** 3
**Contribution:** 1
**Rating:** 2
**Confidence:** 3

**Summary:**

This paper studies LLMs' ability to faithfully simulate a probability distribution, with a focus on the Bernoulli distribution. It is shown that while an LLM can identify the correct Bernoulli distribution from data, it fails to faithfully generate samples from a Bernoulli distribution. To reduce the sampling bias, the paper proposes Verbalized Rejection Sampling (VRS) which prompts an LLM to perform rejection sampling in language. Experiments show that the method leads to significant reduction in the LLM's sampling bias. The paper also provides some theory to prove that VRS outperforms direct sampling under certain conditions.

**Strengths:**

1. The problem of whether or not an LLM can faithfully simulate a distribution is interesting.
2. The paper gives both empirical and theoretical evidence to demonstrate the advantage of VRS over direct sampling.

**Weaknesses:**

1. My main concern is the limited scope of the paper. The empirical and theoretical investigations are almost exclusively focused on the Bernoulli distribution (with some additional results for the binomial distribution in the appendix). The leap from the single-parameter binary Bernoulli distribution to multivariate, continuous, or complex categorical distributions is substantial, and the paper does not provide empirical or theoretical evidence that VRS would be effective in these more complex settings. This renders the paper a specific solution to a very narrow problem. For the sake of sampling from a Bernoulli or binomial distribution, why not just use a standard software?
2. Moreover, as is shown in Section 6.1, the effectiveness of VRS can be heavily impacted by the prompt. In particular, a prompt that instructs the LLM to compute $M$ correctly can even degrade its sampling performance. This instability, combined with the lack of theoretical guarantees, makes VRS feel more like an ad hoc method rather than a robust and general approach.

**Questions:**

1. The paper positions itself as a first step towards studying LLM's probabilistic reasoning skills, and is motivated by practical applications such as agent-based modeling and randomized decision-making. Do the authors think that the insights in this paper (e.g., the knowledge-sampling gap) will carry over to those more practical scenarios? I feel that the nature of stochasticity in this paper's setting and in the more practical settings is quite different. The experiments in this paper are conducted in an isolated "lab" setting, where an LLM is explicitly prompted to generate a sample from a given distribution. But in the more practical settings such as agent-based modeling and randomized decision-making, stochasticity is often implicit (instead of explicitly specified) and embedded in a much larger reasoning chain.

---

> ### Author Response · Authors · 2025-11-28
>
> We thank the reviewer for their thoughtful critique regarding the scope of our study and the practical applicability of Verbalized Rejection Sampling (VRS). We appreciate the opportunity to clarify the motivation behind focusing on fundamental distributions and how these insights translate to broader contexts.
>
> > 1. Scope: Why Bernoulli and Why Not Standard Software?
>
> We chose the Bernoulli distribution precisely because it is the fundamental "atomic unit" of discrete probability. As noted in Section 1, if LLMs cannot faithfully simulate a simple coin flip, they cannot be trusted to sample from more complex distributions. By isolating this failure mode, we provide a clean testbed to analyze the "knowledge-sampling gap".
> Regarding standard software: We fully agree that for engineering tasks where external code execution is available, Python samplers are the gold standard. However, as detailed in Appendix E.1 and E.2, our work targets:
> 1. Tool-Free Environments: Many real-world deployments (e.g., secure environments, chat-based agents) restrict code execution.
> 2. Intrinsic Capability: We study "verbalized computing" to understand if LLMs can simulate stochasticity through reasoning alone.
> 3. Foundation for Complexity: We have demonstrated that the method extends to Binomial distributions in Appendix C. Since many complex distributions can be decomposed into Bernoulli trials, getting the "atom" right is a prerequisite for scaling. Also, the theoretical analysis we provide in Section 6.2 is itself general and applies to any noisy accept/reject sampler, not just LLMs. Additionally, the bound presented in Corollary 1 also generalizes under the assumption that direct sampling yields a distribution within TV distance $c$ of the target. This is a fairly uniform assumption about the error of direct sampling. For a tighter analysis one would need to investigate the structure of the error of direct sampling case-by-case (i.e., distribution-by-distribution) which we consider out-of-scope for the current work.
>
> > 2. Robustness vs. "Ad Hoc" Nature
>
> We respectfully disagree that VRS lacks theoretical guarantees. We believe there is a misunderstanding w.r.t. Section 6.1. We showed that VRS is NOT impacted heavily by the prompt, where the native prompt based direct sampling is. In Section 6.2 and Appendix A, we derive formal bounds (Propositions 1 & 2) that quantify the Total Variation (TV) distance of VRS based on the model’s internal bias. We prove that under mild assumptions (where the proposal $Q$ is sufficiently close to the target), VRS is _guaranteed_ to reduce error compared to direct sampling.
>
> Regarding the "instability" when computing $M$: This finding actually reinforces our theoretical analysis rather than suggesting randomness. Both Section 6.1 and Section 6.2 are to make the point that a prompt-based ad-hoc fix is not reliable, VRS works in a robust way exactly because it does not rely on prompt-based fix, but relies on the algorithmic structure of rejection sampling.
>
> > 3. From "Lab" to Practical Agent Scenarios
>
> We believe these insights are directly transferable. While agent simulations often involve "implicit" stochasticity, this eventually collapses into explicit discrete choices (e.g., "Should the agent go Left or Right?", "Accept or Reject the deal?"). If an LLM cannot succeed in the "lab setting" where the prompt explicitly defines the probability, it is highly unlikely to succeed in implicit settings where the probability must be inferred. Understanding the failure mode in the isolated setting is the necessary first step to fixing it in complex reasoning chains.

---

### Official Review · Reviewer_t5LF · 2025-11-01

**Soundness:** 2
**Presentation:** 3
**Contribution:** 2
**Rating:** 4
**Confidence:** 3

**Summary:**

This paper studies if LLMs can generate faithful random samples from Bernoulli distributions described in natural language. The authors identify a knowledge–sampling gap: models can accurately describe probabilities but fail to sample according to them, often showing bias (typically toward “1”). They propose a language-based adaptation of classical rejection sampling, making  the model verbally reasons about whether to accept or reject a proposed sample. Empirical results across 4 LLMs show rejection substantially reduces sampling bias. Theoretical analysis supports that the improvement arises from the rejection sampling mechanism itself, not just prompt effect

**Strengths:**

The paper tries to look into a fundamental problem and is not incremental though feels like an early stage work.
Comprehensive experiments across multiple models, phrasings, and CoT settings; results consistently demonstrate bias reduction.

**Weaknesses:**

Prior work has indicated that LLM agents despite understanding the notion of probabilities struggle with probability sampling (the paper also cite them) "Do LLMs play dice?" Further more developed by works like a "theory of response sampling" which explains distributional shift in sampling/ deviation from distributional statistics based on some kind of value maximization. I think these together might explain the deviation towards 1 in the experiment.

Also the knowing doing gap of LLM, as knowing a concept and unable to use this in sampling/action is not really new. The 'knowing doing gap' has been discussed in many works, seminal work as "LLMs are Greedy Agents: Effects of RL Fine-tuning on Decision-Making Abilities". I dont see this a new finding though the finding here gives a nicve spin to it.

The paper to my understanding argues the algorithmic process of probabilistically filtering of samples mitigates the LLM's inherent sampling bias. So as a verifier it works but give consistant bias in sampling. I think the paper is missing prior work that explains similar observation across similar task lile exploration where verbalized verifiers improve perfromance.

Also a minor concern on why the study limits to one distribution? and do the authors think this will scale? Can you try 0,1 and 1,0 in prompt to reduce order bias or use random tokens to amke sure the encoded value of 0 and 1 does not come into play?

I think the attempt to understand the underlying sampling pheneominon can be a good contribution thought the findings are a bit early stage but the paper fails to place itself and build on existing framework to explain the observation. Also for wider audiance may be say why using tool to fix the sampling problem is not good enough. I am ready to review my score and accept post rebuttal post rebuttal if the discussion help clarify my understanding of how the authors explain the observation in light of prior work. Thanks you.

**Questions:**

Please see weakness

---

> ### Author Response · Authors · 2025-11-28
>
> We thank the reviewer for their constructive feedback.
>
> > 1. Connection to prior work
>
> We appreciate the reviewer pointing us to the "Greedy Agents" and "Theory of Response Sampling" frameworks. We fully acknowledge that the "knowing-doing" gap is a known phenomenon in LLMs. We do not claim to have discovered this gap; rather, our Section 4 is designed to quantify and visualize this specific discrepancy in the context of probabilistic sampling, and showed that chain-of-thought is not a reliable method for closing the gap, to motivate the need for an algorithmic intervention. We will update our related work section to include the missing reference. We want to highlight that unlike previous approaches that attempt to bridge this gap through fine-tuning or heuristic prompt engineering, VRS provides a mechanism to correct the bias at inference time without requiring model updates or access to internal logits.
>
> > 2. “Can you try 0,1 and 1,0 in prompt to reduce order bias …”
>
> We appreciate this suggestion and would like to clarify that we have already conducted this control experiment in Section 4.2. As detailed in Figure 4(b) and Table 1, we tested four different prompt phrasings to rule out order and framing bias:
> * $P_1$: stating only the probability of 1.
> * $P_0$: stating only the probability of 0.
> * $P_{01}$: stating “probability of 1 is p, and probability of 0 is 1-p”
> * $P_{10}$: stating “probability of 0 is 1-p, and probability of 1 is p”
> Our results show that while balanced descriptions ($P_{10}, P_{01}$) slightly improve calibration over single-sided descriptions, significant bias persists regardless of the phrasing. This confirms that the bias is intrinsic to the model's sampling mechanism and not merely an artifact of token order. And our proposed algorithm VRS is able to significantly reduce the bias regardless of the prompt phrasing.
>
>
> > 3. “Why the study limits to one distribution?”
>
> While the main text focuses on Bernoulli as a foundational "unit test" for sampling, we have demonstrated scaling to more complex distributions in the Appendix. Binomial Distribution: In Appendix C, we extend our theoretical analysis for arbitrary $n$ and empirical evaluation to Binomial distributions ($n \in \{1, 2, 3, 4, 5\}$). Table 5 shows that VRS significantly outperforms direct sampling in reducing Total Variation Distance (STVD) for Binomial distributions as well.
>
> In principle, VRS can be extended to other distributions. Just like in classical rejection sampling, each new distribution requires a tailored derivation of acceptance probabilities. Extending VRS will likely require distribution-specific modification and theoretical analysis, which is feasible, but nontrivial.
>
> We view VRS as a **proof of concept for a broader class of natural-language-based probabilistic methods**. It shows that we can: (1) adapt probabilistic reasoning algorithms to LLMs through language, (2) rigorously analyze their behavior using classical tools, and (3) derive formal guarantees even when the underlying sampler (the LLM) is biased or noisy. In this sense, VRS is algorithmically general.
>
> The theoretical analysis we provide in Section 6.2 is itself general and applies to any noisy accept/reject sampler, not just LLMs. Additionally, the bound presented in Corollary 1 also generalizes under the assumption that direct sampling yields a distribution within TV distance $c$ of the target. This is a fairly uniform assumption about the error of direct sampling. For a tighter analysis one would need to investigate the structure of the error of direct sampling case-by-case (i.e., distribution-by-distribution) which we consider out-of-scope for the current work.
>
>
> > 4. Why not use External Tools?
>
> We agree that for standard engineering applications, calling a Python sampler is the correct approach. However, as discussed in Appendix E.1 ("Why not ask the LLM to call an external sampler?"), addressing the "native" sampling capability is crucial for three reasons:
>
> 1. Tool-Free Environments: LLMs are increasingly deployed as autonomous agents in "text-only" loops where code execution is restricted for safety or architectural reasons.
>
> 2. Stochastic Decision Making: Agents often need to break ties or make randomized decisions (e.g., "explore vs. exploit") purely through internal reasoning. If an LLM cannot "flip a coin" internally, its decision-making becomes deterministically biased, and leads to safety and fairness issues.
>
> 3. Scientific Inquiry: Just as we study whether LLMs can do math without a calculator to understand their reasoning capabilities, studying whether they can simulate randomness reveals fundamental properties of their internal representations and the limitations of alignment.

---

### Note · Authors · 2025-11-28

**Comment:**

We would like to thank the reviewers for their time and effort in reviewing our paper.

**Withdrawal Confirmation:**

I have read and agree with the venue's withdrawal policy on behalf of myself and my co-authors.